# Transcriptional decomposition reveals active chromatin architectures and cell specific regulatory interactions

Sarah Rennie[1], Maria Dalby[1], Lucas van Duin[1] & Robin Andersson [1]

Transcriptional regulation is tightly coupled with chromosomal positioning and three-dimensional chromatin architecture. However, it is unclear what proportion of transcriptional activity is reflecting such organisation, how much can be informed by RNA expression alone and how this impacts disease. Here, we develop a computational transcriptional decomposition approach separating the proportion of expression associated with genome organisation from independent effects not directly related to genomic positioning. We show that positionally attributable expression accounts for a considerable proportion of total levels and is highly informative of topological associating domain activities and organisation, revealing boundaries and chromatin compartments. Furthermore, expression data alone accurately predict individual enhancer–promoter interactions, drawing features from expression strength, stabilities, insulation and distance. We characterise predictions in 76 human cell types, observing extensive sharing of domains, yet highly cell-type-specific enhancer–promoter interactions and strong enrichments in relevant trait-associated variants. Overall, our work demonstrates a close relationship between transcription and chromatin architecture.

[1] The Bioinformatics Centre, Department of Biology, University of Copenhagen, Ole Maaloes Vej 5, DK-2200 Copenhagen, Denmark. Correspondence and requests for materials should be addressed to R.A. (email: robin@binf.ku.dk)

The three-dimensional organisation of a genome within a nucleus is strongly associated with cell-specific transcriptional activity[1,2]. On a global level, transcriptional activation or repression is often accompanied by nuclear relocation of chromatin in a cell-type-specific manner, forming chromatin compartments[3] of coordinated gene transcription[4–6] or silencing[7]. Locally, chromatin forms sub-mega base pair domains of self-contained chromatin proximity, commonly referred to as topologically associating domains (TADs)[8,9]. TADs frequently encompass interactions between regulatory elements, such as between promoters and enhancers[10–13], as well as between co-regulated genes[9], which reflects cell-type-restricted transcriptional programmes[14]. In contrast, ubiquitously expressed promoters are enriched close to domain boundaries[8] and co-localise in the nucleus[15]. A tight coupling between transcriptional activity and chromosomal positioning is further supported by positional clustering of co-expressed eukaryotic genes[16,17], a phenomenon that is preserved across taxa[18]. In addition, neighbouring gene expression correlation co-evolves and is particularly evident at distances below a mega base pair[19]. Furthermore, facultative heterochromatin may cover mega base pair regions[20] and thus affect several neighbouring genes. These observations are in line with coordinated gene expression and chromatin states within TADs[9,21], suggesting that the coupling between gene expression and chromatin architecture is, at least partly, linked to chromosomal positioning.

Genetic disruption or chromosomal rearrangements of TAD boundaries may result in aberrant gene transcription as a consequence of altered regulatory activities[22] or regional chromatin states[23]. While this suggests a key role for chromatin structure in correct transcriptional activity, a general directional cause and effect between chromatin architecture and transcriptional activity is unclear. On one hand, TAD boundaries are to a large extent invariant between cell types[8,9] and have been suggested to be mainly formed by architectural proteins[11,24–26] independent of transcription[15]. On the other hand, transcription seems to play a major role in the maintenance of regulatory organisation within TADs[15,27]. In addition, chromatin compartments are not affected by architectural protein depletion[26]. Rather, nuclear three-dimensional co-localisation of genes seems to be driven to a large degree by transcriptional activity[6,15,28–30].

The strong relationships between transcription, chromatin state, chromosomal positioning and three-dimensional chromatin organisation can be exploited for inference of one feature from another. Compartments of transcriptional activity can be deduced from genome-wide chromatin interaction data[3,11] and, inversely, co-expression is indicative of enhancer–promoter (EP) interactions[31]. In addition, RNA expression was placed among the top-ranked features for predicting EP interactions in a recent machine-learning approach[32]. However, it is unclear what proportion of expression is associated with chromatin topology and chromosomal positioning and what proportion is reflecting regulatory programmes independent of the former. A way to systematically extract and separate these components from expression data could lead to new insights into chromatin topology and cell-type-specific transcriptional regulation. This is of high importance since genome-wide profiling of chromatin interactions using chromatin conformation capture techniques such as HiC[33] is largely intractable due to several limitations including low spatial resolution, high cost and requirements of large abundant cellular material.

In this study, we establish a transcriptional decomposition approach to investigate and model the coupling between transcriptional activity, chromosomal positioning and chromatin architecture. Via modelling of expression similarities between neighbouring genomic loci, we show that the proportion of expression related to genome architecture can be separated from effects independent of genomic positioning, and that both components are strongly represented with contributions depending on cell type and location. We demonstrate that the positionally dependent (PD) component is highly reflective of chromatin organisation, revealing chromatin compartments and structures of transcriptionally active TADs. We further demonstrate how transcriptional components can be used to infer cell-type-specific chromatin interactions and find informative transcriptional features, including enhancer expression strength, which distinguish long-distance interacting from non-interacting EP pairs. We demonstrate the accuracy of our approach in well-established cell lines and then decompose expression data from 76 human cell types in order to investigate their active chromatin architectures. Our results indicate extensive sharing of expression-associated domain structures across human cells, reminiscent of that observed for active TADs. Promoter-localised, positionally independent (PI) events as well as EP interactions are, on the other hand, highly cell-type specific. Finally, we demonstrate how transcriptional components and predicted EP interactions can be used to gain insights into the genetic consequences of complex diseases. We observe variable enrichments of genetic variants in expression components across cell types and traits and find several EP interactions in which disease-associated SNPs at enhancers may cause aberrant expression of distal genes. Taken together, we observe a tight coupling between transcriptional activity and three-dimensional chromatin architecture and suggest that regulatory topologies, domain structures and their activities may be inferred by RNA expression data and chromosomal position alone.

## Results

**Transcriptional decomposition of RNA expression data.** We posited that a proportion of steady-state RNA expression is reflecting three-dimensional chromatin organisation. We reasoned that a transcription unit (TU) is likely to be more similar in terms of expression to its proximal TUs than to distal loci, which are likely to be associated with different domains of chromatin interactions. Thus, coordination of expression is related between positional and chromatin neighbourhoods (Fig. 1a). However, expression is also influenced by gene-specific transcriptional and post-transcriptional regulatory programmes independent of a TU's chromosomal position. Therefore, in order to investigate the coupling between transcriptional activity and genome organisation, we need to be able to estimate the proportion of expression from a genomic region that can be explained by its chromosomal position. To this end, we developed a transcriptional decomposition approach (Fig. 1b) to separate the component of expression reflecting an underlying positional relationship between neighbouring genomic regions (PD) component) from the expression dictated by TUs' individual regulatory programs (PI component). Our strategy is based on approximate Bayesian modelling[34] and utilises replicate measurements to decompose normalised aggregated RNA expression read counts in genomic bins into two components (PD and PI) and some constant intercept (Methods). The expression associated with each genomic bin was modelled as a sum of the two transcriptional components (random effects model), as illustrated in Fig. 1a. The decomposition of this sum into its respective components (summands) can be interpreted as an optimisation problem over a chromosome to best separate the proportion of total expression that is similar between consecutive genomic bins (PD) from position-independent expression (PI).

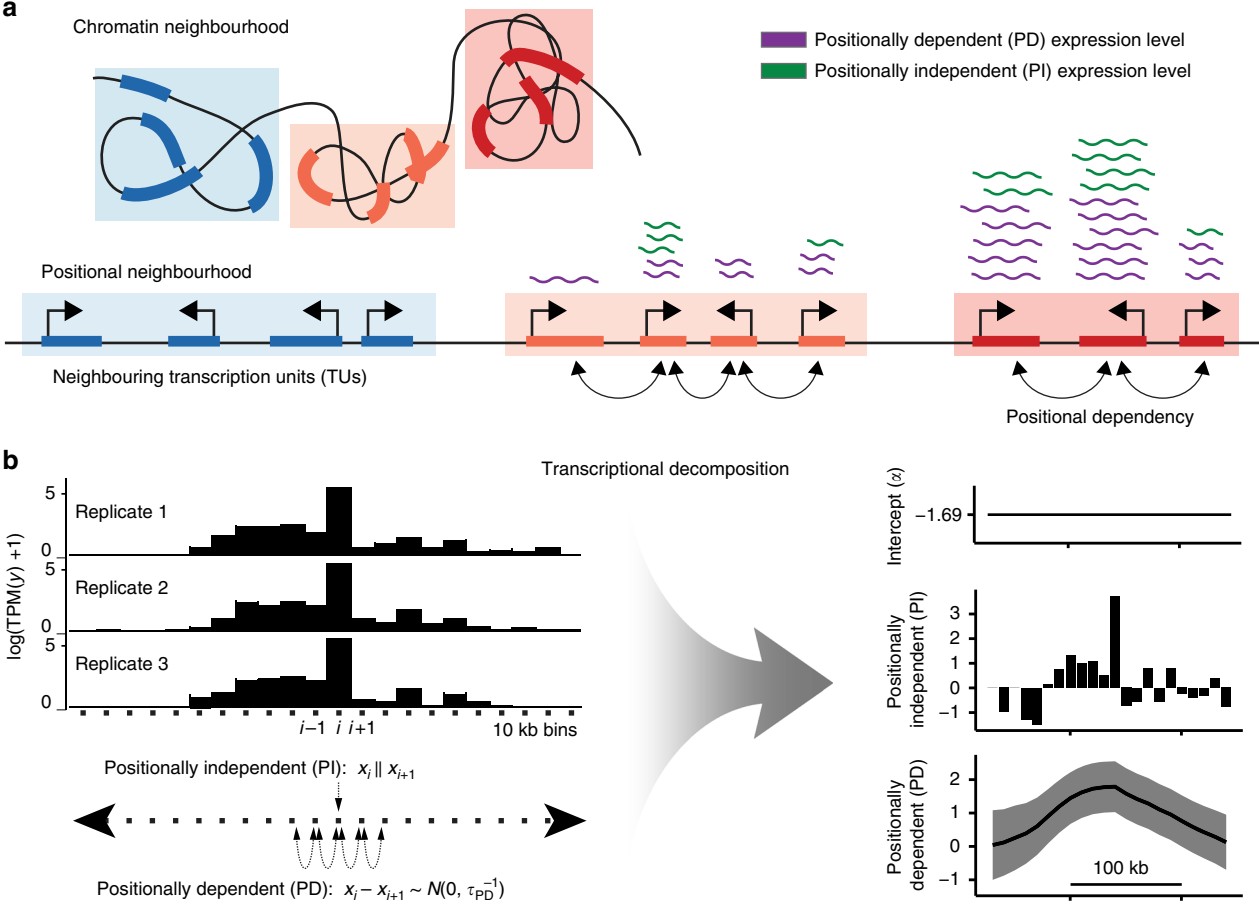

**Fig. 1** Transcriptional decomposition separates the proportion of RNA expression related to chromosomal position from positionally independent (PI) effects. **a** Schematic illustrating how RNA expression derives from two major sources. The positionally dependent (PD) component reflects the underlying dependency between linearly proximal TUs in chromosomal, positional neighbourhoods, which are related to chromatin neighbourhoods of TU three-dimensional proximity. The PI component reflects localised, gene-specific regulatory programs unaffected by the positioning of TUs. **b** Overall strategy of how replicated samples are decomposed into transcriptional components. Via approximate Bayesian modelling, normalised RNA expression count data quantified in genomic bins (here 10 kb), are decomposed into an intercept ($\alpha$), a PI component and a PD component. The PD component is modelled as a first-order random walk, in which the difference between consecutive bins is assumed to be Normal and centred at 0 (Methods). The variable $y$ represents the expression level, $x$ represents the component value in bin $i$ and $\tau$ represents the precision of a normally distributed random variable

**Decomposition reveals positional dependency on expression.** We applied transcriptional decomposition to replicated cap analysis of gene expression (CAGE)[35] data[36], measuring transcription initiation sites and steady-state abundances of capped RNAs from GM12878, HeLa-S3 and HepG2 cells. In the modelling, we used 10-kb genomic bins to capture large-scale positional effects. We first asked whether the information content of the two components contained different biological interpretations as posited. Upon close inspection (Fig. 2a), the PD component displayed considerably broader patterns than the PI component and appeared to be highly similar between cell types. Despite the overall similarities in the PD component, we identified large differences in individual loci between cells, as exemplified by *KCNN3* and *NOS1AP* genes (Fig. 2b, c). For instance, *NOS1AP* is in GM12878 cells associated with low PD signal and appears to reside in polycomb-repressed chromatin, as indicated by high levels of histone modification H3K27me3 and low levels of histone modification H3K36me3. HepG2 and HeLa-S3 cells displayed opposing signals for this locus, suggesting that the PD component contains information about chromatin compartments.

In order to understand the observed effects on a genome-wide scale, we compared the PD component with HiC-predicted chromatin compartments[11] in GM12878 cells. We observed that the PD signal in regions of active chromatin was higher than in those of facultative or constitutive chromatin, while constitutive chromatin states had the lowest signal (Fig. 3a). Congruently, we noted that regions of positive PD signal were highly enriched in expressed genes within a given cell type (odds ratio ranging from 4.3 to 15.9, $p<2.210^{-16}$ for all three cell types, Fisher's exact test), with an average of 93% of expressed genes and 70% of genes overall located in positive PD regions. Overlaying the PD component on HiC compartment boundaries also showed clear shifts, which were many magnitudes stronger than what could be detected using the PI component (Supplementary Fig. 1a). In addition, we observed that the PD component clearly correlated within compartments more strongly than between compartments (Supplementary Fig. 1b-e). These results show that the states and boundaries of compartments are reflected by positionally attributable expression (PD signal) and its relative change between consecutive bins.

To examine the localised patterns observed for expression levels not attributable to position (PI component, Fig. 2a), we trained a random forest model (Methods) on GM12878 transcriptional components to predict the presence or absence of DNase I hypersensitive sites (DHSs), histone variant H2AZ

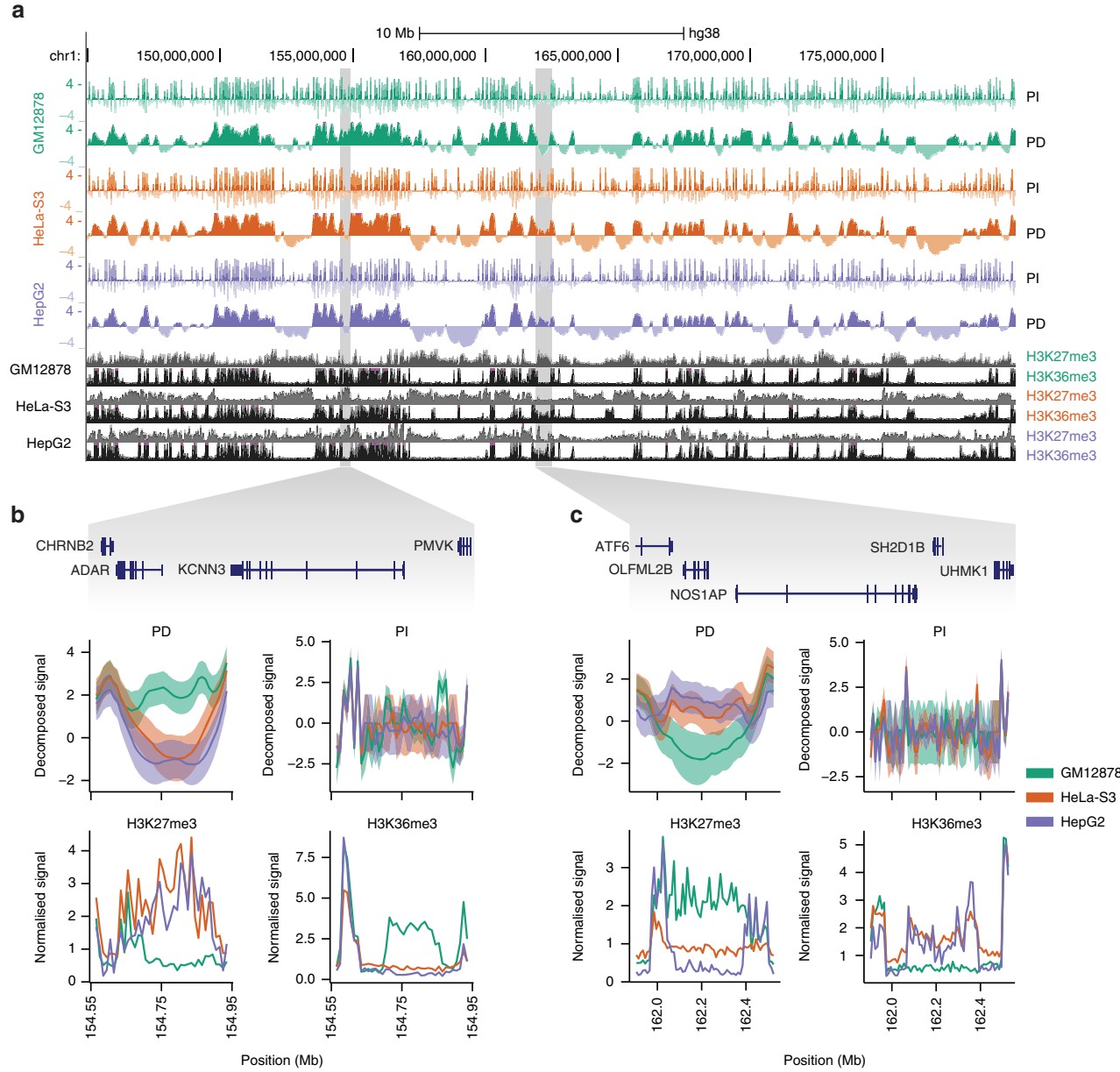

**Fig. 2** Transcriptional decomposition across chromosomes. **a** PI and PD components (mean ± standard deviations), as well as H3K27me3 and H3K36me3 ChIP-seq data for GM12878, HeLa-S3 and HepG2 cells at locus chr1:145,000,000–180,000,000. **b, c** Loci (highlighted in **a**) around *KCNN3* (**b**) and *NOS1AP* (**c**) genes showing cell-type-specific PD signals. The PD and PI signals and ChIP-seq data associated with repression (H3K27me3) and activation (H3K36me3) are shown

and post-translational histone modifications H3K4me3 and H3K27ac (binarised DNase-seq and ChIP-seq data[37] in each bin), each associated with features of (transcriptionally) active regulatory elements[38]. The resulting models allowed us to generate a probability distribution for each mark given each transcriptional component. For all tested marks, we observed a clear bias with stronger predictive power from the PI component than the PD component (Fig. 3b). These results indicate that the PI component, in contrast to the PD component, carries information about regulatory element-localised and expression-level-associated effects.

Overall, we found that both the PD and PI components explained considerable proportions of expression levels in GM12878 cells (Supplementary Fig. 2a). Each component on a median explained roughly half of the expression levels in expressed bins, with contributions varying between loci

(Supplementary Fig. 2b). When compared with HeLa-S3 cells, we observed clear differences in both the PD and PI components between cell types and that these differences were orthogonal between components (Fig. 3c, d). Differentially expressed (DE) bins in the PD component (Benjamini–Hochberg FDR < 0.01, z-score approximations of the posterior estimates of the PD differences between GM12878 and HeLa-S3 cells, Supplementary Data 1) were in line with its relationship with chromatin compartments, to a large degree associated with cell-type differences in chromatin states, changing from silent to H3K36me3-associated active chromatin in upregulated bins (Supplementary Fig. 2c, d). The PI component, on the other hand, showed localised differential expression of bins (Supplementary Data 2) that were associated with cell-type-specific enrichments of predicted binding sites from sequence motifs recognised by expressed transcription factors (TFs) (Fig. 3e and

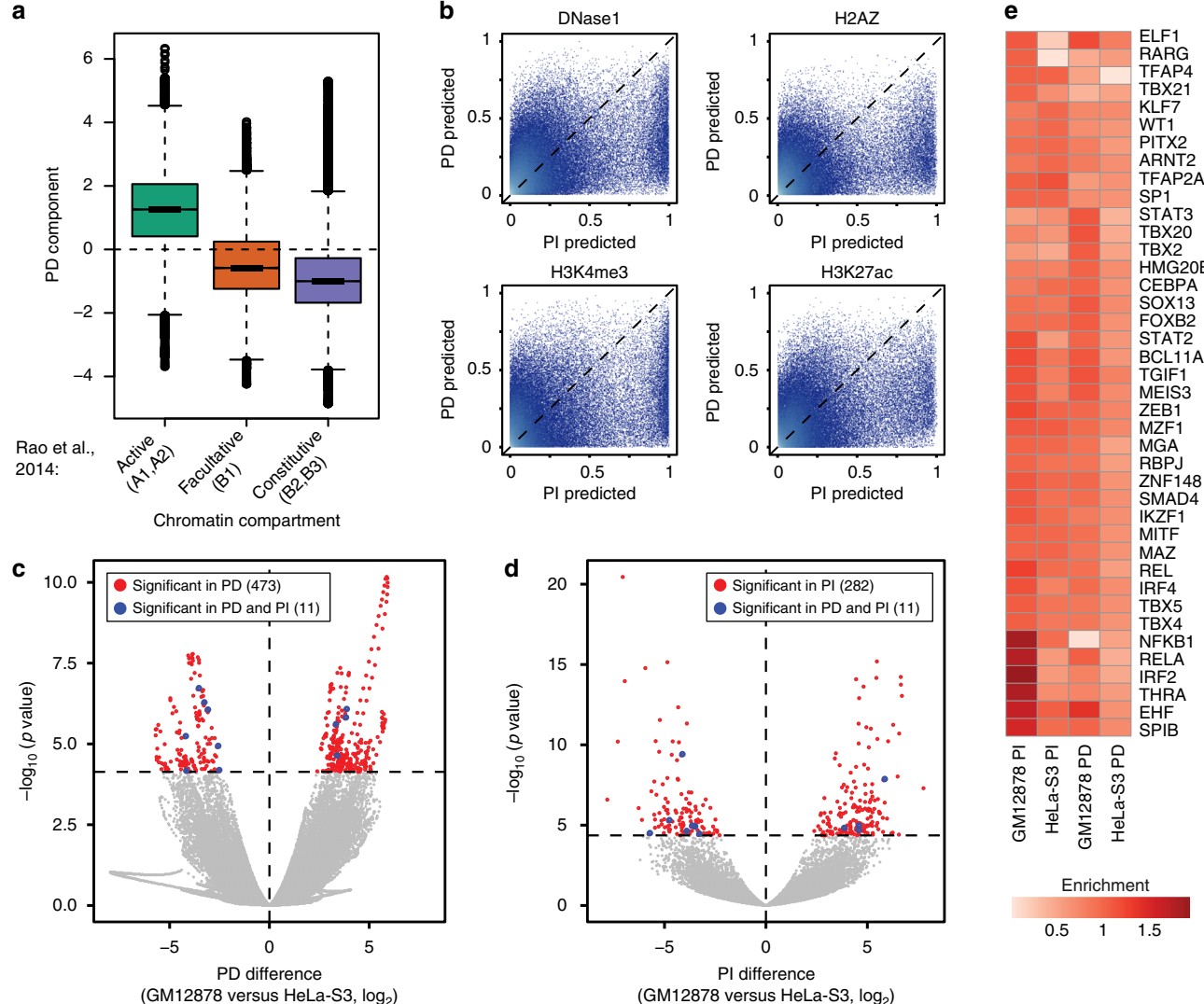

**Fig. 3** Transcriptional components reveal chromatin compartments and localised regulatory element-associated effects. **a** Box-and-whisker plot of GM12878 PD signal grouped according to HiC-derived chromatin compartments. The lower and upper hinges of boxes correspond to the first and third quartiles of data, respectively, and the whiskers extend to the largest and smallest data points no further away than 1.5 times the interquartile range. **b** Random forest class (presence/absence) probability of DNase1, H2AZ, H3K4me3 and H3K27ac as learned from the PI component (x axes) and the PD component (y axes). **c** PD difference (x axis, difference in PD value for log-transformed expression) versus false discovery rate (FDR)-adjusted p-value (rescaled by −log$_{10}$) for PD component differential expression between GM12878 and HeLa-S3. Red represents significant bins unique to the PD component (corresponding to 473 bins), and blue represents those common to both (11 bins). **d** As **c** but for PI component (red: 282 significant bins unique to the PI component). **e** Expressed TF motif enrichment around expressed CAGE-derived promoters associated with GM12878 or HeLa-S3-biased differentially expressed PD or PI bins, versus all expressed CAGE-derived promoters. See Supplementary Figure 2e for all enriched expressed TFs

Supplementary Fig. 2e), for instance NFKB and IRF in GM12878 cells.

Similar patterns of transcriptional components derived from GM12878 CAGE data were found when we applied transcriptional decomposition, guided by CAGE-estimated hyperparameters, to GM12878 RNA-seq data[38] (Supplementary Fig. 3, Supplementary Table 1, Methods), confirming the observed characteristics and differences between CAGE-derived components. Taken together, we conclude that expression data can be decomposed into a PD component, revealing chromatin compartment activity, and a PI component carrying information about localised, independent expression-associated events.

**Expression-associated domains mark active chromatin topology.** Apart from displaying clear shifts at compartment boundaries, we noted that the PD component contained sub-patterns

within broader consecutive regions of positive signals (Fig. 2a). We posited that such structures could represent finer, transcriptionally active chromatin organisation not necessarily reflecting chromatin compartments, but rather boundaries of active TADs. To test this hypothesis, we trained a generalised linear model (GLM) to predict HiC-derived TAD boundaries[11] from features derived from transcriptional components (Supplementary Table 2, Methods). The GLM yielded an area under the receiver operating characteristic (ROC) curve (AUC) of 0.73 (AUC 0.85 in regions of positive PD signal), indicating that there is information in expression data to infer TAD boundaries (Supplementary Fig. 4a, b). Among the features considered, the model ranked the PD gradient (first-order derivative), the PD inter-cell stability and PD variance as among the most important for predicting TAD boundaries (Supplementary Fig. 4c). Based on the GLM feature importance ranking, we devised a score to rank

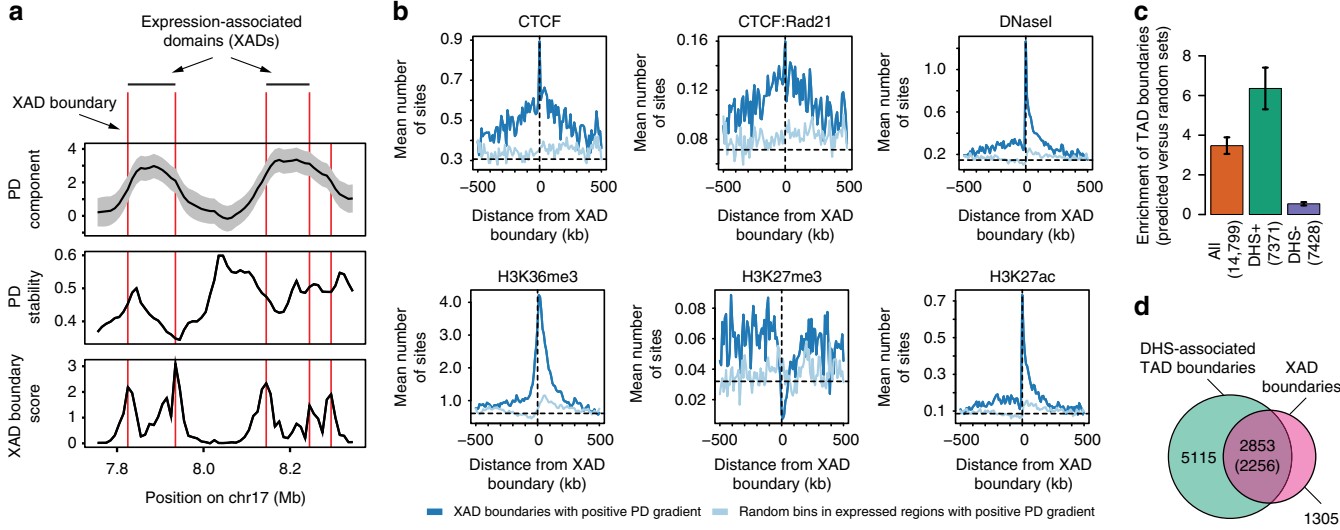

**Fig. 4** Expression-associated domains mark regions of active topological domains. **a** Approach for identifying boundaries of expression-associated domains (XADs) based on a PD boundary score. PD signal (mean ± standard deviations), PD stability (across cell PD standard deviation) and the XAD boundary score are shown. **b** Average GM12878 profiles of binarised ChIP-seq data for CTCF, CTCF in combination with Rad21 (cohesin), DNaseI, H3K36me3, H3K27me3 and H3K27ac at XAD boundaries with positive PD gradient (dark blue) and at random expressed bins with positive PD gradient (light blue). The vertical dotted line represents boundary locations and the horizontal dotted line represents background mean for a given mark. **c** Enrichment of GM12878 TAD boundaries among XAD boundaries compared to random bins proximal to expressed bins (DHS+ for DHS associated, DHS− for DHS non-associated). Error bars were derived from generating the random bin set 100 times and calculating the standard deviation of their enrichment versus the actual boundaries. **d** Venn diagram of association between GM12878 XAD boundaries and proximal (within five bins) DHS-associated TAD boundaries

PD boundaries at significant gradients in the PD signal that also had low positional standard deviation (high stability) across cells (Fig. 4a, Methods). In GM12878 cells, we detected 4158 boundary locations of PD sub-patterns. We refer to the regions demarcated by PD boundaries as expression-associated domains (XADs, Fig. 4a). In general, GM12878 XAD boundaries coincided with, or were very close to, the locations of TAD boundaries (Fig. 4d).

We next assessed the occurrence of DHSs, ChIP-seq binding site peaks for architectural proteins CTCF and Rad21 (a subunit of cohesin) and histone modifications H3K36me3, H3K27me3 and H3K27ac around GM12878 XAD boundaries (Fig. 4b). We observed an enrichment of DHSs, H3K27ac and binding of CTCF alone and, albeit weaker, in combination with Rad21 around XAD boundaries. In addition, H3K36me3, H3K27ac and DHSs were more enriched downstream than upstream of positive-gradient XAD boundaries. The opposite trend was observed for H3K27me3. The observed DHS and ChIP-seq patterns around XAD boundaries (Fig. 4b) resemble those around TAD boundaries in active compartments[24,39–41]. In line with these and the GLM results, we observed that GM12878 XAD boundaries were in general highly enriched in HiC-derived TAD boundaries[11] from the same cells (>3-fold enrichment over the background, Fig. 4c). Specifically, XAD boundaries were highly enriched in DHS-associated TAD boundaries (>6-fold enrichment over the background), but not in those distal to DHSs (Fig. 4c and Supplementary Fig. 5a, b). Out of 4158 GM12878 XAD boundaries, 2853 (69%) were proximal (within five bins) to DHS-positive TAD boundaries (Fig. 4d). Still, 69% (5115 out of 7371) of DHS-positive TAD boundaries were distal to XAD boundaries, suggesting that XADs represent a subset of DHS-positive TADs. Incomplete overlap between XAD and TAD boundaries is expected since the detection of XAD boundaries relies on transcriptional activity. Indeed, joint XAD and TAD boundaries were primarily found within active chromatin, while XAD-unsupported DHS-positive TAD boundaries more frequently resided in facultative or constituitive heterochromatin (Supplementary Fig. 5c, $p < 1 \times 10^{-258}$, $\chi^2$ test). Furthermore, joint

XAD and TAD boundaries displayed a higher HiC[11] chromatin interaction directionality[8] than DHS-positive TAD boundaries distal to XAD boundaries (Supplementary Fig. 5d, e), indicating that expression-associated chromatin is linked with stronger TAD boundaries (greater insulation). However, both sets were similarly associated with Rad21 (Supplementary Fig. 5f, g), whose co-binding with CTCF is believed to provide strong TAD boundaries[24,39,40]. Taken together, these results demonstrate that expression data (PD component) can be used to infer chromatin topology in active chromatin compartments.

**Decomposed expression data predict chromatin interactions.** Since positionally attributable RNA expression was strongly associated with structures of transcriptionally active TADs, we questioned the utility of expression data alone in reflecting individual proximity-based interactions. Namely, what does it mean to be proximal in the nucleus, from a transcriptional perspective?

To this end, we collected a total of 25 features in GM12878 that may be derived from expression data sets alone (Supplementary Table 3), relating to the values, differences, cross-cell-type correlations, standard deviations and stabilities of the transcriptional components, as well as XAD boundary insulation and features relating to local CAGE-measured promoter and enhancer activities, directionality scores and their chromosomal distances. To test the power of expression-universal features to classify chromatin interactions, we used a random forest classification scheme where we compared it to GM12878 promoter-capture HiC (CHiC) data[12] (Methods). To enable a direct assessment of the relationships between transcriptional component-derived features and chromatin interactions, we mapped CHiC data at 10-kb resolution. We used a lenient interaction score threshold to define positively interacting bin–bin pairs (bin–bin distance >50 kb and ≤2 Mb, based on CHiCAGO[42] score ≥ 3, see Supplementary Table 4), for which each pair referred to a CHiC promoter bait and a potential target that overlapped with a transcribed promoter[36] or transcribed enhancer[31,43]. In order to

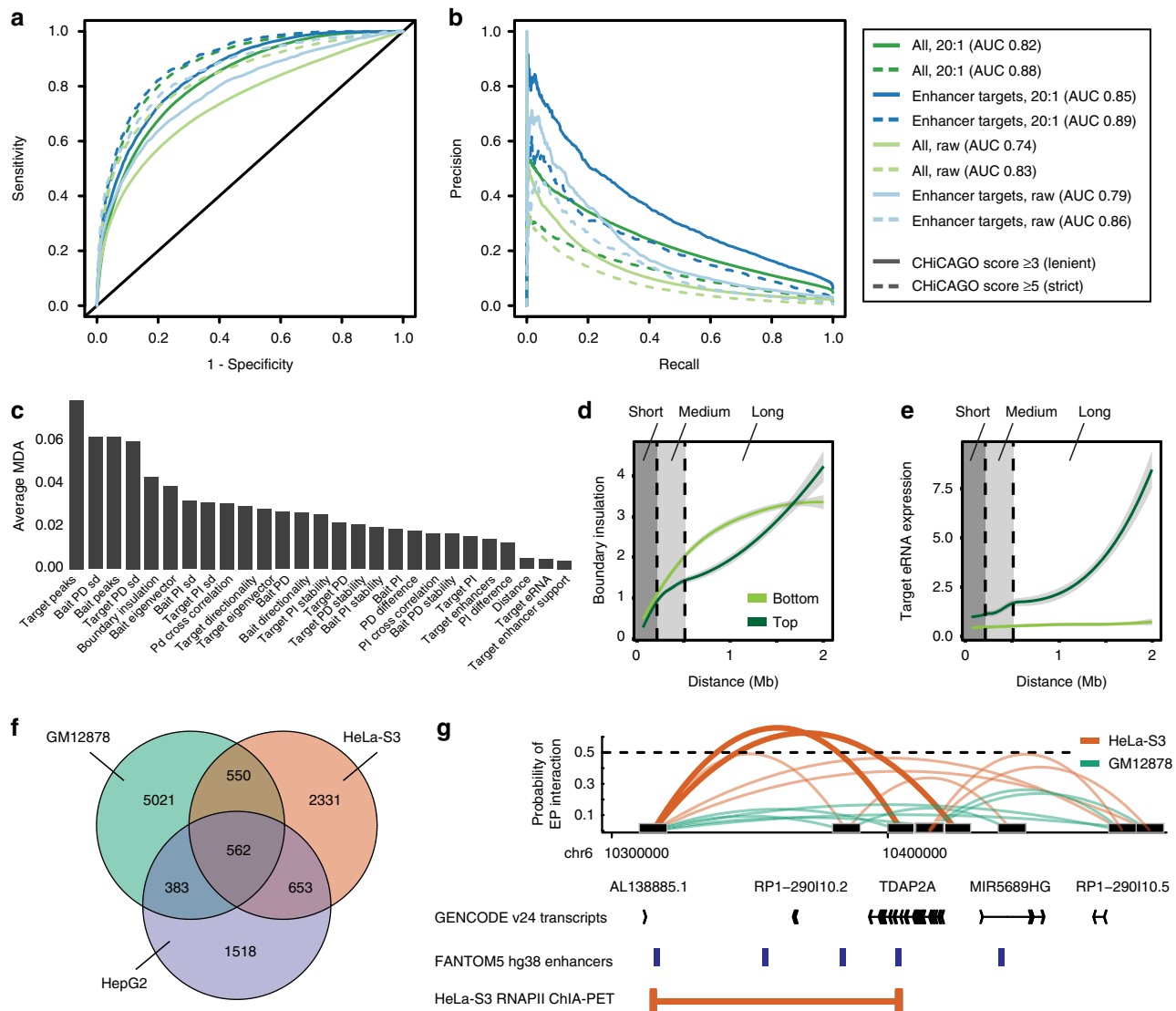

**Fig. 5** Expression data are predictive of cell-type-specific regulatory interactions. **a**, **b** Performance curves for predicting bait–target interactions from CAGE-universal features, split according to CHICAGO score, testing negative-to-positive ratios and target feature type. **c** Features predictive of bait–target interactions, ordered by average mean decrease accuracy (MDA) across models from 10-fold cross-validation. **d**, **e** Loess curves representing feature separation over distance between high (top) and low (bottom) predicted probabilities, shown for features (**d**) XAD boundary insulation (*nbounds*, number of XAD boundaries between bins) and (**e**) enhancer expression at the target (*eRNA_targ*, mean tags per million across three replicates). **f** Overlaps of predicted bait–enhancer interactions between GM12878, HeLa-S3 and HepG2 cells. **g** An example of a loop predicted in HeLa-S3, but not in GM12878 cells, validated by HeLa-S3 RNAPII ChIA-PET interaction data

deal with the resulting unbalanced data, we over-sampled[44] the positives and under-sampled the negatives in the training data to a fully balanced set across distances in a 10-fold cross-validation scheme. However, validations were performed using a 20:1 negative:positive (NP) ratio, similarly to a previous method[32], but also on raw (unmodified) NP ratio. Thus, in each cross-validation round, we fully balanced the training data and predicted on held-out data at 20:1 or raw NP ratios.

Overall, we observed a good predictive performance (AUC:0.74) on raw NP ratios for lenient interaction thresholds (Fig. 5a). Using a strict interaction threshold (CHiCAGO score ≥ 5) for evaluation increased the AUC (0.83) but reduced the recall (Fig. 5b). At a 20:1 NP test ratio, the overall predictive performances increased (AUC 0.82 and 0.88 for lenient and strict interaction thresholds, respectively). Interestingly, we observed a better precision and recall when evaluation of results was limited to enhancer targets

(Supplementary Data 3, AUC of 0.85 and 0.89 for lenient and strict interaction thresholds at a 20:1 NP ratio, respectively; see Supplementary Fig. 6 for the effect of interaction thresholds on EP prediction performance). We observed varying predictive performances across chromosomes (Supplementary Fig. 7), likely reflecting differences in gene densities and transcriptional activities. Despite the overall good EP predictive performance, both precision and recall decreased over increasing distances on held-out test data, in accordance with an increasing NP ratio (Supplementary Fig. 8). To circumvent the distance effect, we established a distance-dependent threshold in random forest voting, guided by the optimal F1 score, significantly improving the predictive performance over distance (Supplementary Fig. 9). Taken together, we conclude that there is a wealth of information from properties of expression data alone which could explain chromatin interactions, and in particular EP interactions. This

suggests a tight coupling between expression of regulatory elements and their proximity.

Next, we asked what properties of expression data explained chromatin interactions. High transcriptional activity at the target or bait was ranked among the top features for predicting chromatin interactions (Fig. 5c). In addition, the standard deviation of the PD component at either the bait or the target was informative (Fig. 5c). Predicted interactions had a lower PD standard deviation on average (Supplementary Fig. 10), suggesting that high-confidence interactions are associated with stable estimates of active chromatin compartments. Boundary insulation, defined as the number of XAD boundaries predicted between the bait and the target, also ranked highly, with a weaker insulation associated with a higher probability of a chromatin interaction compared to other pairs at a similar distance (Supplementary Fig. 10). Since the observation of this property forms the definition of TAD domains, this lends support of XAD boundaries as predictors of boundaries of transcriptionally active TADs. Separately training three random forest classifiers for short-, medium-, and long-range distances (covering bait–target distances within (50,200), (200,500) and (500,2000) kb, respectively) allowed us to further investigate features driving long-range interactions (Supplementary Fig. 11a). As expected, boundary insulation had a higher influence on long-distance interactions than shorter ones (Fig. 5d). Interestingly, when we specifically considered interactions between enhancers and promoters, eRNA expression at the target enhancer clearly distinguished predicted positive from predicted negative interactions, and its importance increased over increasing distances (Fig. 5e).

Motivated by the good performance in predicting EP interactions, we next used the GM12878-trained random forest model to predict EP interactions also in HeLa-S3 and HepG2 cells (Supplementary Data 4, 5). We noted that many predicted interactions were specific to each cell type (Fig. 5f, 48–78%), with only a small fraction (9–20%) of interactions shared between all three cell lines (see Supplementary Fig. 12 for results using a strict interaction threshold). For instance, the promoter of gene *TDAP2A* was predicted to interact with an ~100-kb downstream enhancer specifically in HeLa-S3 cells, supported by HeLa-S3 RNAPII ChIA-PET interaction data[38] (Fig. 5g).

In support of eRNA production at positive interactions, both enhancers and promoters in predicted cell-type-specific interactions clearly showed an expression bias towards the cell type in which the interactions were identified, in contrast to shared EP interactions (Supplementary Fig. 11b, c). These results indicate that expression of regulatory elements is reflecting both their cell-type-specific regulatory activities and their regulatory interactions. This is supported by observations that regulatory active enhancers that are interacting with promoters are more likely to be transcribed than non-interacting ones[45]. Taken together, we demonstrate that transcription is highly informative of three-dimensional chromatin architecture and may be used to accurately infer EP interactions.

**Transcriptional decomposition reveals cell-type differences**. We have demonstrated that positionally attributable expression (PD component) can be used to accurately infer boundaries of active TADs, as well as differences in chromatin compartments and EP interactions between well-established and biologically distal cell lines. We continued with exploring what insights could be gained by transcriptional decomposition of primary cell expression data, for cell types for which chromatin topology is to a large degree unknown. We applied transcriptional decomposition to replicated primary cell CAGE data[36] expanding to a total of 76 human decomposed cell types from 249 CAGE libraries

(Supplementary Data 6, Methods). Using the resulting components, we calculated XAD boundaries and extended the previous set of pairs defined in the GM12878 training above to a common set of bin pairs across all cell types (Supplementary Table 5), over which EP interactions were predicted.

Consistent with observations in cell lines (Fig. 2a), the PD and PI components displayed opposing trends when compared across cell types. Many genomic bins showed a highly similar PD signal across cells, while the PI component was rarely shared across more than a few cell types (Supplementary Fig. 13a,b). The PI component was grouped more closely according to cell-type association than the PD component and clearly distinguished blood cells from mesenchymal, muscle and epithelial cells (Fig. 6a, b). XAD boundaries tended to be either highly cell-type specific or ubiquitous (Supplementary Fig. 13c), similar to that of TAD boundaries[46], reflecting the cell-type-specific behaviour of features around XAD boundaries (Fig. 4b), e.g. DHSs[47]. In comparison, EP interactions exhibited very little agreement across cell types (Fig. 6c), with very few shared interactions across the full repertoire of samples (Supplementary Fig. 13d). Interestingly, cell-type-specific EP interactions were on average more distal than interactions shared across cells (Supplementary Fig. 13e). When examining EP interactions between groups of cells, we observed clear differences at key identity genes. Leukocyte-biased genes *ARHGAP25* (Fig. 6d) and *CD48* (Supplementary Fig. 14) showed clear differences in predicted EP interactions, which were validated by GM12878 CHiC interaction data. Taken together, these results indicate largely invariant chromatin compartments between cells and that cell-type identities are governed by differences in EP interactions or position-independent effects, which may be driven by localised, cell-type-specific TF-binding events.

**Components guide interpretation of trait-associated variants**. Many complex genetic diseases are associated with genetic variants outside of protein-coding gene sequences in gene-distal enhancers[31,48] and these variants have been suggested to explain a large portion of disease heredity, in particular in immunological disorders[49]. Thus, we investigated the utility of decomposed expression data and predicted EP interactions in the interpretation of disease-associated genetic variants. Based upon enrichment analysis of trait-associated SNPs[50] and those in strong linkage disequilibrium (Methods), we observed largely variable preferences between traits and transcriptional components (Supplementary Fig. 15a). Interestingly, several diseases and traits showed a preference for enrichment in positionally attributable expression (PD-positive bins) or within two bins of XAD boundaries, indicating that associated genetic variants may alter chromatin compartments or the activities or structures of enclosing TADs. Furthermore, while most trait enrichments were restricted to a few cell types, many of those biased to positionally attributable expression, for instance Crohn's disease, displayed broad association across the panel of investigated cell types (Supplementary Fig. 15b). Monocytes, including those subjected to various pathogens, T cells, B cells and natural killer cells were found amongst the most highly ranked cell types associated with Crohn's disease in PD and PI components as well as at XAD boundaries (Fig. 7a). This is consistent with the strong links between Crohn's disease, inflammation, innate and adaptive immune system deficiency[51]. Amongst genes in SNP-associated bins, we found Crohn's disease-associated genes *STAT3*, *ATG16L1* and MHC genes *HLA-DWB*, *HLA-DRA* and *HLA-DQA2*[52]. Similar to Crohn's disease, lymphoid leukaemia was strongly associated with cells of the immune system (Supplementary Fig. 16a), but had different genes associated with

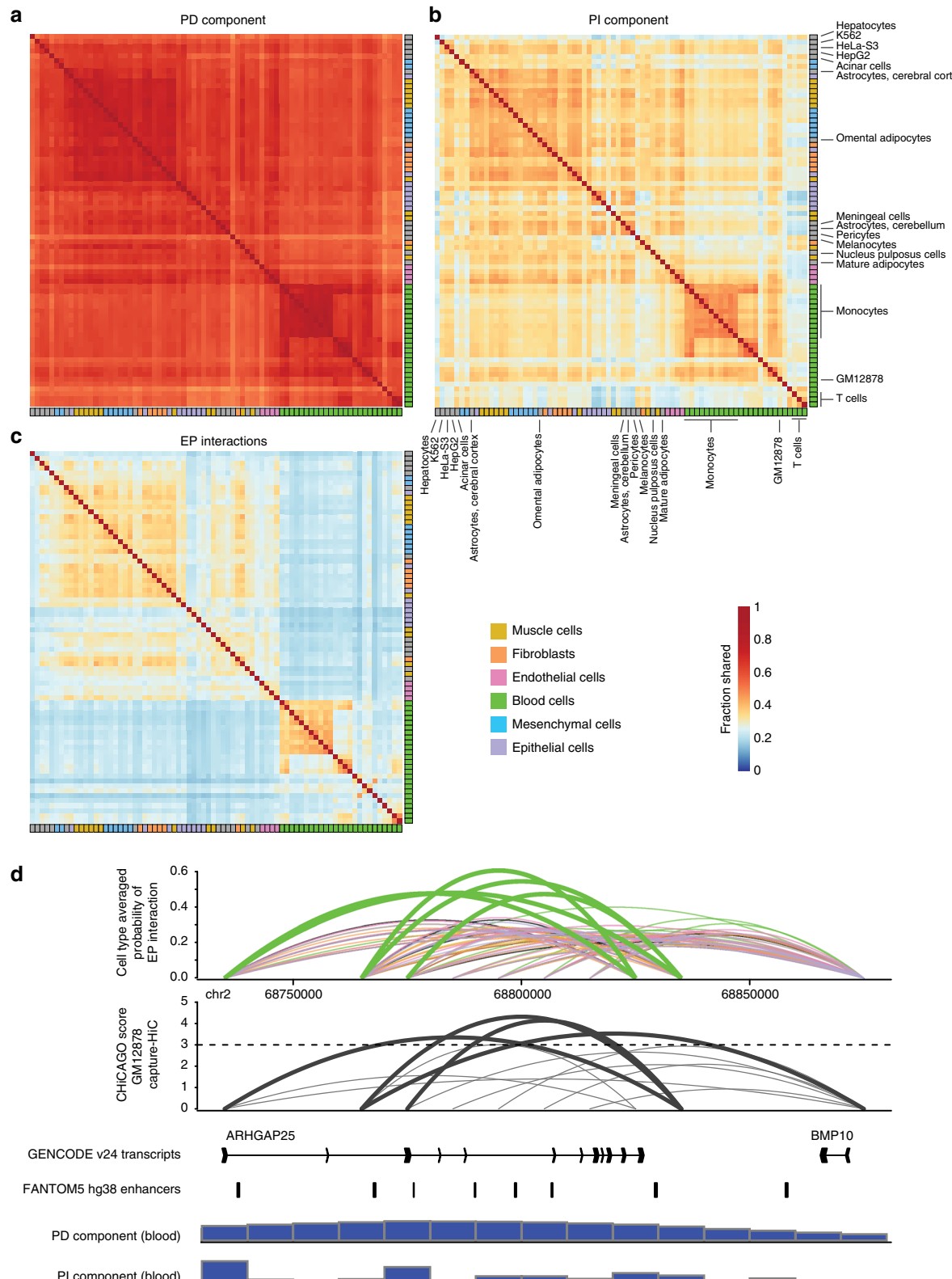

**Fig. 6** Transcriptional decomposition across 76 human cell types. **a–c** Heat maps depicting pairwise similarities of the PD (**a**), PI (**b**) components and EP interactions (**c**) across cell types. All similarity scores were calculated between cell types using $1-L_1$ norm on binary data sets based on the sign of the PD, PI components or the presence/absence of EP interactions. Cell-type ordering is fixed according to the results of a hierarchical clustering (complete linkage) of the raw expression data. **d** Predicted probability of EP interactions averaged across groups of cells and CHiCAGO score of interaction based on GM12878 capture HiC data. GENCODE v24 transcripts, FANTOM5 enhancers and the average PD and PI component across blood cells are displayed below

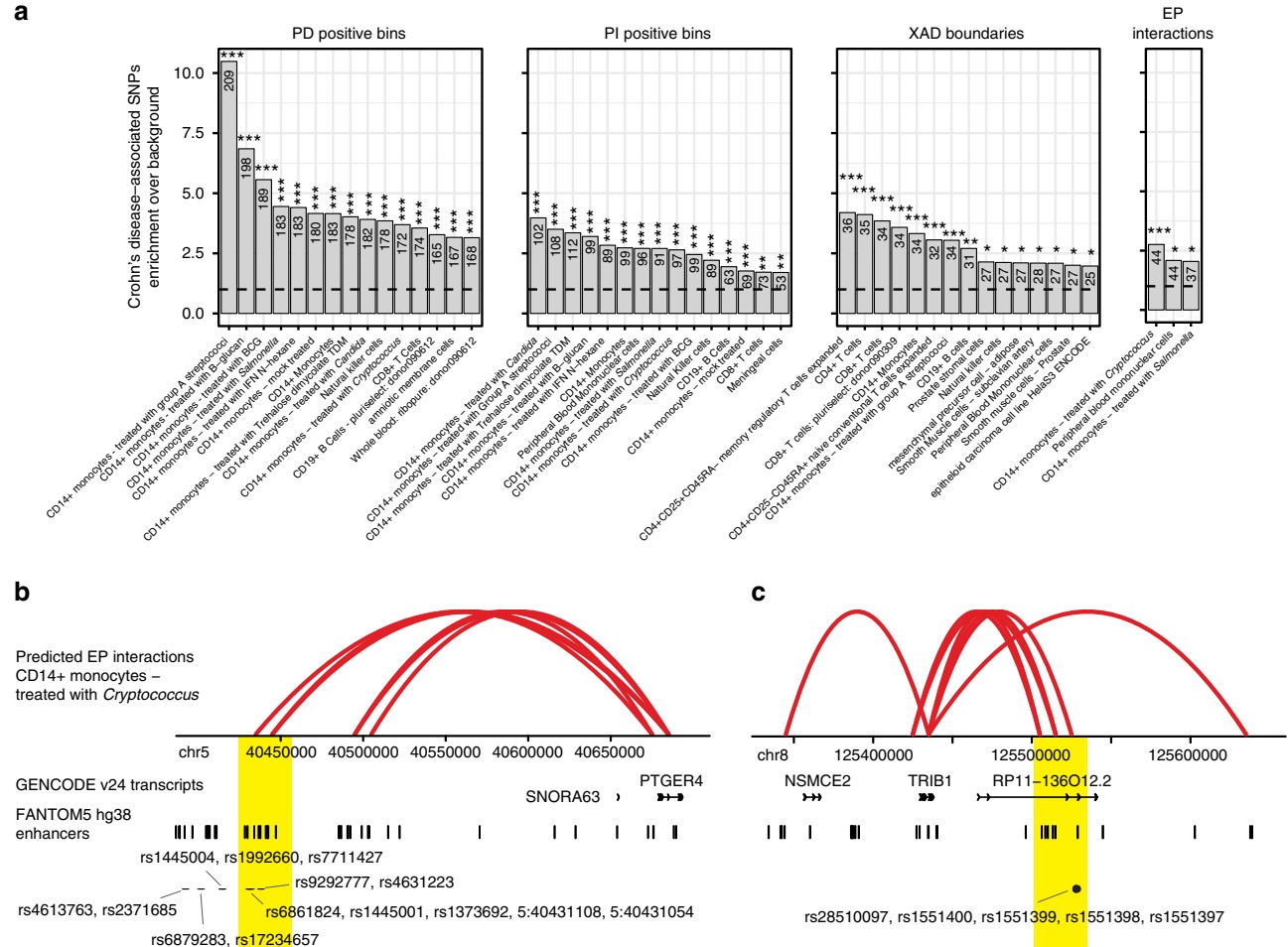

**Fig. 7** Analysis of Crohn's-disease associated SNPs reveals cell-type preferences and regulatory associations. **a** Crohn's disease SNP enrichments in transcriptional components, XAD boundaries and inferred enhancer–promoter interactions (FDR-corrected $\chi^2$ tests based on the PD/PI positive bins or the presence of XAD boundaries/target enhancers per cell type, and trait-associated SNPs). See also Supplementary Figure 16 for enrichments in lymphoid leukaemia. Significance stars above bars are interpreted as: *$P<0.1$, **$P<0.01$ or ***$P<0.001$. **b**, **c** Predicted EP interactions in CD14+ monocytes treated with *Cryptococcus*, for genes *PTGER4* (**b**) and *TRIB1* (**c**) for which interacting enhancers overlap or are in close proximity with disease-associated SNPs (highlighted in yellow). Predicted EP interactions, GENCODE v24 transcripts, FANTOM5 enhancers and the locations of relevant SNPs are displayed below

the PD and PI components and at XAD boundaries, including *IRF4*, *IKZF1*, *ARID5B*, *CEBPE*, *IRF8* and *GCSAML*.

For Crohn's disease, we found a significant enrichment of enhancer-overlapping SNPs in predicted EP interactions of monocytes treated with *Cryptococcus* or *Salmonella*, as well as in peripheral blood mononuclear cells (Fig. 7a). SNP-associated enhancers were predicted to interact with several disease-relevant genes, including *PTGER4* (Fig. 7b), *TRIB1* (Fig. 7c), *CCL1*, *OTUD1* and *ARMC3*[52,53], and genes so far not associated with Crohn's disease *ENKUR* and *THNSL1*. For example, promoters of *PTGER4* were predicted to interact with enhancers more than 250 kb upstream of the gene situated in an LD block of GWAS-associated SNPs (Fig. 7b). Similarly, promoters of *TRIB1* were predicted to interact with an ~100-kb downstream enhancer overlapping a disease risk locus of SNPs (Fig. 7c). These results clearly demonstrate how transcriptional decomposition of expression data can be used to gain insights into disease aetiology, and thus the value of our generated resource.

## Discussion

A detailed understanding of the intricate relationship between transcription and three-dimensional chromatin organisation and to what extent they are attributable to each other has been lacking. Under the assumption that the link between transcriptional activity and chromatin organisation is reflected, at least partly, by chromosomal positioning, in this study we have aimed to separate the fraction of RNA expression that is associated with chromatin topology from localised, independent effects. To this end, we have developed a transcriptional decomposition approach that decomposes expression data into positionally attributable (PD) and independent (PI) components along chromosomes. We show that positionally attributable expression, according to the PD component, closely reflects chromatin compartments and domain architecture, and accounts for a sizeable overall fraction of TU expression levels. This points to the existence of large constraints on genomic organisation, suggesting that the global maintenance of chromatin formation is crucial for correct and precise transcriptional output. On the other hand, the PI component carries information about regulatory element-localised and expression-level-associated effects.

We suggest that PI effects may be the result of cell-type-specific programmes involving, for instance, the differential binding or effect of TFs, potentially based on a layer of localised regulation not necessarily attributable to higher-order three-dimensional chromatin. However, we observe that the level of positional dependency of expression data varies between loci and cell types, suggesting locus- and cell-dependent effects of topological organisation and activity, as well as varying the impact of individual gene regulatory programmes. This suggests that genes strongly associated with non-positional expression may be more resistant to perturbations in chromatin domains, such as deletions of TAD boundaries, particularly in the context of a certain disease.

The usage of expression data to infer topological chromatin organisation is limited by the inability to inform on transcriptionally silenced, closed or poised states. However, by focussing only on expressed TUs, such as those observed with CAGE, and their relative relationships, we can attempt to understand patterns that are highly relevant to various cell types of interest. For example, the majority of detected XAD boundaries were proximal to TAD boundaries associated with an open chromatin site in GM12878 cells. Interestingly, XAD boundary locations were seen to be strongly informed by the presence of a stable PD signal across cell types, reflected in their high degree of sharing, which closely corresponds to the nature of TADs, whose locations appear to be largely cell-type invariant[8,46]. This is likely connected to the enrichment of promoters for actively transcribed housekeeping genes at the locations of TAD boundaries[8,15], whose expression levels remain stable across cell types. Our predictions also suggest that active transcription at open chromatin loci is linked with stronger TAD boundaries, according to the observation that chromatin interaction directionality scores were stronger at TAD boundaries proximal to XAD boundaries compared to those distal from XAD boundaries, which were associated with closed chromatin. These results suggest that the presence of active transcription aids in the reinforcement of the insulation properties associated with boundary formation, but may also point to different regulatory mechanisms or roles involved in boundaries according to their link to active transcription.

We have shown that expression data alone are predictive of fine-scale chromatin interactions, based on predictive models incorporating expression component information, expression strength, stability and distance. Our results concur with, and significantly extend, the previous finding that CAGE is an informative predictor of interactions[31,32]. Crucially, we bring up the question of what proximity within the nucleus means from the standpoint of transcriptional initiation, and our results suggest the two to be tightly coupled. Analysis of predicted features points to a model whereby a strongly expressed TU with stable positionally attributable expression is the key predictor of within-domain interactions in a given cell type, and whereby strength of target expression becomes more important for predictions over long distances. Interestingly, of all predicted chromatin interactions, those restricted to EP interactions showed the strongest predictive power, were biased towards the cell types in which they are actively transcribed and target enhancer RNA expression levels rapidly increased with the distance between the bait and the target.

Since transcriptional decomposition is broadly scalable across large numbers of samples and applicable to different expression-sequencing assays, including CAGE and RNA-seq, it potentially paves the way for large-scale cost- and time-effective computational analyses across atlases of high-quality expression data sets, such as FANTOM5[36] or GTEx[54]. We have applied our models to 76 cell types from the FANTOM5 consortium, generating predicted components, XAD boundaries and EP interactions. This

resource allows a deeper understanding of the dynamic regulation at key identity genes in a wide diversity of cellular states not yet subjected to methods informing on their higher-order structures. Comparison across cell types revealed extensive sharing of positionally attributable expression, while PI effects and, in particular EP interactions, were to a large degree cell-type specific, particularly at long distances, and separated cells into groups of similar type. These results indicate that chromatin compartments are mostly invariant across cells and that fine-tuning of cell-type-specific expression levels is mainly mediated by promoter-localised, position-independent effects or enhancer regulation. The differences in cell-type specificity also indicate that positionally attributable expression, on its own, is not sufficient to inform on EP interactions. While positionally attributable expression seems to reflect the overall large-scale chromatin state in which a TU resides, and informs on the architectural compatibility between regulatory elements, many EP interactions are cell-type specific and associated with punctuated elevated enhancer and promoter expression levels. These results argue for RNAPII-centred chromatin architectures[55,56], whereby transcriptional regulation is associated with cell-type-specific tethering of EP pairs to RNAPII foci, leading to coordinated expression increases.

While many complex diseases have clear cell-type-specific effects and that positionally attributable expression was chiefly shared between cells, we found an unexpected preference for enrichments of trait-associated genetic variants in the PD component. This suggests that the aetiology of certain diseases may be coupled with the disruption or alteration of TAD structures or association with chromatin compartments, which finds support in the literature[22,57]. Other traits were associated with cell-type-specific enrichments of variants in the PI component or at enhancers in cell-type-specific EP interactions, likely to cause altered gene expression levels. Our generated resource of regulatory interactions enabled us to identify several cases of predicted EP interactions for which enhancer overlap with disease-associated SNPs was linked with promoters of genes often associated with the disease itself. Overall, we expect that our transcriptional decomposition approach and resource will have large implications for future interpretations of genetic variants associated with disease in cell types that are otherwise largely intractable.

## Methods

**Processing of CAGE data sets**. CAGE data were produced by the FANTOM5 project[36]. The data mapped to hg38 and CTSSs (CAGE transcription start sites) were clustered into tag clusters (TCs) according to decomposition peak identification (DPI) generation as per FANTOM5[58]. Only samples with more than 0.5 million tags mapping within the TCs were included in the analysis.

Enhancers were called based on bidirectional balanced RNA signatures as per the FANTOM5 consortium[31]. Enhancers were only identified distal to known exons (±100-bp region from boundaries) and transcription start sites ±300 bp defined by GENCODE v24 annotation. In total, 63,285 enhancers were identified across 1829 libraries. Due to varying noise levels across CAGE libraries and the intrinsic low expression levels of transcribed enhancers, library-specific noise levels were estimated to define a robust set of active (transcribed) enhancers in each sample. For each library, expression was quantified in randomly sampled mappable genomic regions distal to assembly gaps, DNase hypersensitive sites (ENCODE), known exons and gene TSSs (GENCODE) to create a genomic background expression distribution. For each library, we called an enhancer active (used) if its expression was above the 99.9th quantile of the library's genomic background expression distribution. The resulting robust set consists of 60,215 enhancers over 1829 libraries, being significantly expressed above the background in at least one library. The expression was quantified and TPM normalised according to the total number of mapped reads within the full set of TCs (tags per million (TPM)).

CTSS files containing positions of raw CAGE counts from libraries for the ENCODE tier 1 cell lines (GM12878, K562, HeLa-S3 and HepG2) were intersected with non-overlapping 10-kb regions across the genome. For each chromosome (chr1–chr22 and chrX), regions were defined from coordinate 1 (1-based) and in consecutive complete 10-kb blocks, up to two blocks after the last bin containing a single CAGE tag across the set of libraries. Region sizes of 40 kb and 100 kb were

also considered, but 10 kb was exclusively used in the analyses for comparability to high-resolution chromatin capture data sets.

Due to the sparsity of CAGE tags in eterochromatic regions and regions poorly mapped, including the case of libraries with poor sequencing depth, a zero-inflated negative binomial distribution was assumed to hold for the counts across the 10-kb bins on each chromosome, given by, for bin $i$, $Y_i \sim \text{ZINB}(p_i, \mu_i, k)$, where $Y_i$ is the bin count, $p_i$ represents the zero-probability parameter, $\mu_i$ represents the mean and $k$ the size or dispersion parameter. The mean is given by $\mu_i = k\frac{1-p_i}{p_i}$, with variance $\sigma_i^2 = \mu_i(1 + \frac{\mu}{k})$. The model assumes a zero-truncated negative binomial distribution on the non-zero counts ($Y_i \sim \text{NB}(\mu_i, k)$ for $y_i = 1, 2, \dots$) with probability equal to $1 - p_i$.

**Transcriptional decomposition of CAGE data.** The transcriptional decomposition models the mean log count as a combination of an intercept and two random effects, set up as $\nu_i = \alpha + \text{PD}_i + \text{PI}_i$ where $a$ is the intercept, $\text{PD}_i$ and $\text{PI}_i$ are random effects for the PD and PI components, respectively and $\nu_i$ is the linear predictor given by $\nu_i = \log(\mu_i) - \log(E)$ for library depth $E$ where $\log(E)$ is the offset term. The PD component is modelled as a first-order random walk $\text{PD}_i - \text{PD}_{i+1} \sim \text{N}(0, \tau_{\text{rw}}^{-1})$ where $\text{PD}_i - \text{PD}_{i+1}$ represents the component difference between successive bins and $\tau_{\text{rw}}$ is the precision of the normally distributed differences, with mean 0. The PI component assumes that bins are represented by a vector of independent and Gaussian-distributed random variables with precision $\tau_{\text{iid}}$.

The model was fit in the form of a Bayesian mixed model using the R[59] package R-INLA[34] which implicitly assumes a Gaussian field on the parameter space and uses a Laplacian approximation to allow for fast and deterministic convergence of parameters. Hyperparameters were defined for the size and zero-probability parameters (gamma-distributed and Gaussian-distributed prior distributions, respectively) of the negative binomial distribution, and precision parameters $\tau_{\text{rw}}$ and $\tau_{\text{iid}}$ (log-gamma-distributed priors) for the PD and PI components, respectively. Three replicates for each of the ENCODE tier 1 cell lines were included, assuming library depths equal to the sum of the tags in their respective CTSS files. ENCODE cell lines were modelled concurrently for each chromosome, assuming a common distribution for the hyperparameters.

In order to allow for efficient prior estimations, we scaled the random walk components such that the average variance (measured by the diagonal of the generalised inverse) is equal to 1. In order to achieve efficient convergence and avoid precisely defining priors beforehand, the option diagonal = 1 was set within the INLA call (to avoid falling into sparse errors). The posteriors based on the converged model were then fed into a second model specifying diagonal = 0 in order to achieve more accurate estimates.

**Transcriptional decomposition of RNA-seq data.** To see if chromosomal transcriptional decomposition is broadly applicable to RNA data sets, we applied the same modelling procedure to deeply sequenced GM12878 RNA-seq samples (https://www.encodeproject.org/experiments/ENCSR843RJV/)[38]. The libraries were mapped using hg38 using HISAT[60] with default parameters. Multimappers were removed using samtools[61] and reads were binned at 10-kb resolution (based on bins defined previously for CAGE) using bamCoverage[62]. Transcriptional decomposition was applied to the resulting counts, assuming library depth to be the total of the genome-wide bin counts. A range of hyper-parameters for the PD and PI components was tested (corresponding to the CAGE-defined values of the parameters, and −3 to +3 relative) and the combination was selected, per chromosome, whereby the PD component correlated the most strongly with the PD component in GM12878 CAGE (ignoring regions not within 25 bins of a TU in RNA-seq).

**Analysis of differences between PD and PI components.** To address the proportions of total RNA expression levels allocated to each of the PD and PI components, the raw CAGE counts in GM12878 were used to identify 10-kb bin regions with a mean count greater than 50 tags across the three replicates, thus ensuring that most selected bins were positive in both components (but removing those which were not). For each of these bins, the fraction $\frac{\text{PD}}{\text{PD}+\text{PI}}$ was calculated, where PD and PI are the respective estimates of the PD and PI components in the bin. These fractions were plotted as a histogram and the median value was identified as an average ballpark for relative allocation to PD component versus the PI component.

Genes, according to the GENCODE v24 annotations, were deemed to be expressed in GM12878 cells if they were associated with at least five tags at their TSS across the three GM12878 CAGE libraries and compared to the sign of the PD signal in GM12878 for their containing bins. The full set of genes was based on those associated with at least one FANTOM5 hg38 library.

Compartment coordinates for GM12878 cells[11] were lifted over from hg19 to hg38 using liftOver tool with default settings. For simplicity of interpretation, the five compartment types from Rao et al.[11] were associated with active (A1,A2), facultative (B1) and constitutive (B2,B3) chromatin environments. Overlaps between the 10-kb bin regions and compartment regions were used to assign bins to chromatin environments, removing bins not having a corresponding overlap (genomic overlaps were identified using the R package GenomicRanges[63]). Bins representing shifts between two different chromatin environments were deemed to

be boundary bins and for sets corresponding to each possible shift (e.g. active to constitutive), the mean of the PD component signal was calculated for bins at an increasing distance on either side of the boundary bins, in steps of up to 10 kb and including 500 kb. Cases at a given distance whereby another boundary bin was encountered were removed. For each set, a random sampling was used to generate a list of random boundary bins of length equal to the set size. Background sets were each generated 100 times to form distributions.

Intra-chromosomal bin pairs which overlapped with an annotated compartment were assigned an integer according to how many compartment boundary bins (see above) were between them (boundary insulation). For all bin pairs, both the absolute first-order difference in the PD signal and the correlation between the four ENCODE cell lines were calculated. The differences and the correlations were averaged using the median either at each distance from 10 kb apart to 2 Mb apart or across all distances, separately for each possible boundary insulation.

ChIP-seq and DNaseI-seq GRCh38 bigwig and peak data were downloaded from Ensembl FTP and Ensembl biomart based on Ensembl regulatory evidence v 84[37]. For each mark (DNase1, H2AZ, H3K4me3 and H3K27ac), 10-kb bin regions were overlapped with binding locations to give the presence or absence of the mark in each bin. For each of the PI and PD components, the bin estimate, first-order difference between bin estimates, standard deviation of the bin estimate and stability of the bin estimate (standard deviation across cell-type PD component standard deviations) were calculated. For the two components separately, a random forest model was trained with the mark presence or absence as the response. Out of the bag probabilities, for the models based on the PD data and the PI data were compared directly between the two components.

Posterior estimates and standard deviations of the linear combinations representing the difference in the PD or PI component between GM12878 and HeLa-S3 equivalent bins were generated from the transcriptional decomposition models (described above). Since posterior distributions of the estimates are approximately Gaussian, approximate $p$ values from $z$-scores were generated in order to produce standardised scores for the differences. A Benjamini–Hochberg correction was applied according to the number of bins containing an active TU (such that there were ≥10 tags across the three replicates in at least one of HeLa-S3 or GM12878), and using an FDR < 0.01 to generate a list of significant DE bins in each component. H3K27me3 and H3K36me3 ENCODE Broad Institute bigwig data (from Ensembl Regulatory Build v 84) were quantified in 10-kb genomic bins. The aggregated signal values in each bin were TPM normalised (according to all genomic bins). The TPM values for H3K27me3 and H3K36me3 were then inspected at DE bins between GM12878 and HeLa-S3 cells.

Active regions within the DE PD and PI bins were tested for TF-binding enrichment using RTFBSDB[64]. Active regions were defined as expressed FANTOM5 DPI TCs[58], requiring >1 TPM in at least two of the six libraries (three GM12878 and three HeLa-S3). DPI TCs were extended to regions of −500/+100 around TSSs. DE regions of the PI components with upregulation in HeLa-S3 or in GM12878 (HeLa-S3 PI, GM12878 PI), and DE regions of the PI components with upregulation in HeLa-S3 or in GM12878 (HeLa-S3 PD, GM12878 PD), were tested individually against the universe of expressed TC regions. The analyses were performed using the human database of TF-binding motifs imported from the Cis-BP database[65], restricted to motifs recognised by TFs expressed in the six libraries of interest, measured using functional data profiling gene expression levels as specified in RTFBSDB. Significantly enriched known motifs (FDR < 0.0001) were selected in each of the four tests and plotted together in a heatmap using the R package pheatmap (version 1.0.8) with clustering of TFs based on Euclidean distance.

**XAD boundary predictions.** TAD regions based on 1-kb resolution HIC data in GM12878 were downloaded[11] and lifted over to hg38, requiring a 1:1 correspondence between regions defined in each of the two builds. Boundaries were assigned to 10-kb bins based on overlaps. Adjacent bins with boundaries were dealt with by assigning the boundary to the bin with the largest overlap, so that no two adjacent bins contained boundaries, resulting in a total of 14,799 distinct bin-sized regions containing TAD boundaries.

Features were generated per bin according to those listed in Supplementary Table 2. The set of 10-kb bins containing HIC boundaries (see above) was extended either side by one bin to supply boundary regions to predict on. Due to the lack of CAGE information in non-TSS associated regions, the set of bins was reduced to regions with a potential for a boundary to be predicted by using only the set within 250 kb of a bin containing five tags or more in GM12878 (replicate sum). The response (presence or absence of a TAD boundary with a bin) was generated in two ways, first for all bins within this set, and second, under the requirement that the boundary had to be in a positive random walk region.

In order to assess the features which might distinguish bins containing or not containing boundaries, a logistic regression model was fit using the glmnet[66] package in R. The predictors were scaled before modelling for generating scores of relative importance, using the caret package (version 6.0-73) in R. To test the performance of the model at predicting boundary regions, a 2-fold cross-validation was applied, where the data were randomly split into two equal-sized parts and the total performance was assessed according to the combined predictions on the halves of the data which were held out of the modelling after the corresponding half had been trained on. ROC and precision-recall statistics were generated on a

per-chromosomal basis, and for all of the chromosomes together, using the ROCR package[67] in R and plotted using custom functions.

The top three features from the generalised linear model outlined in the previous section were selected, namely the PD component (PD), the difference in the PD component ($PD_{diff}$) and the PD component stability ($PD_{stab}$). Based on these three features, the following algorithm was implemented for the detection of XAD boundaries:

1. Calculate $X = \frac{PD|PD_{diff}|}{PD_{stab}}$.
2. Calculate the local maxima of $X$ and rank in order of largest to smallest values of $X$.
3. For each chromosome $k$, calculate the proportion of bins of positive PD, ($p_k$) and take the top $N_k$ of the ranked maxima such that $N_k = (p_k / \sum_{chr} p_k)N$, where $N$ is the total target number of XAD boundaries.
4. Split the boundaries into 'up jumps' and 'down jumps' according to positive and negative values of $PD_{diff}$, respectively.
5. Shift the 'up jumps' to the right by one bin (to account for the discrepancy in which of the bins on either end of the $PD_{diff}$ should be called the bound). Leave the 'down jumps' as is.
6. Return the vector sort (down jumps, up jumps).
7. Repeat the above for the ENCODE cell lines.
8. For the final set of bounds in GM12878, choose the set such that the bound is in GM12878 and in at least one other ENCODE cell line (and similarly for the set in the other cell types).

The above algorithm was applied to the ENCODE random walks, using the EMD[68] package in R to generate the local maxima and specifying a target of 5000 XAD boundaries. This resulted in a final list of 4158 XAD boundaries after the final filtering step.

XAD boundaries were calculated at bins where the PD component had either a positive or negative change (first-order difference in PD component). To generate enrichment of ChIP-seq marks around XAD boundaries, only those with a positive change were considered, in order to avoid biases from signal averaging. The results for the negative-gradient boundaries were similar or opposing to those of the positive-gradient boundaries.

For each of CTCF, DNaseI, H3K36me4, H3K27me3 and H3K27ac, the list of 10-kb bins was overlapped with significantly bound sites to determine how many sites appeared in each bin. The mean number of sites overlapping the bins containing the predicted (positive change) XAD boundaries was calculated, then for bins one away from the boundary and so forth in steps of 10 kb up to a distance of 500 kb. Cases at a given distance whereby another boundary bin was encountered were removed to avoid contaminated signal. For the background set, a set of random XAD boundaries of equal cardinality to the real boundaries were generated, under the conditions of a positive gradient and within the set of bins plausible for being predicted as boundaries as previously defined. The same analysis was performed for CTCF:Rad21, which was based on the number of CTCF sites multiplied by the number of Rad21 sites which overlapped a given bin.

To calculate the overlap between the XAD set and the TAD set, regions around the XAD set were extended by five bins (50 kb) on either side and then overlapped with the set of 10-kb regions deemed to contain TAD boundaries. The number of TAD boundaries associated with a DHS which fell within these regions were then counted, together with the number of predicted boundaries which fell in the vicinity of a DHS-associated TAD boundary bin (defined as at least one DHS overlapping the TAD boundary bin or one or more of the two adjacent bins).

To calculate the enrichment of XAD boundaries at the locations of boundary bins overlapping TAD boundaries, the proportion of bins with predicted boundaries which overlapped the TAD boundary bin set was calculated and divided by the same value generated from a random set of boundaries (of the same length as the XAD set, falling within the set of plausible bins). The randomisation step was repeated 100 times and the mean enrichment and standard deviation were calculated. The same analysis was repeated according to where the TAD boundary was DHS associated and where the TAD boundary was not associated with a DHS site (defined in the same way as the overlaps above).

Processed intra-chromosomal HIC data for GM12878 at 10-kb resolution[11] were downloaded and normalised, according to supplied recommendations, using the KR method. The Hi-C directionality score[8] was implemented and applied to locations of TAD boundaries whose corresponding liftovers in hg38 were either supported or not supported by XAD boundaries (within ±5 bins of the boundary). The directionality score was calculated for ±25 bins around the location of the boundary bins and for a jump size of 50 bins to calculate the direction bias of interactions. Means were plotted to obtain patterns of global directionality bias.

**Modelling of chromatin interactions**. Capture HIC data for GM12878 were downloaded and fastq files were extracted using the SRA toolkit. The data were mapped and corrected using HICUPS[69], specifying bowtie2[70] for the mapping and GRCh38 for the genome (downloaded from ftp://ftp.ensembl.org/pub/release-85/). CHiCAGO[42] was then applied to the remapped BAM files to call significant interactions against each bait, using default settings except for specifying 10-kb binning and with the baits defined as the 10-kb bin containing the defined sequences from the Capture HIC protocol[12] lifted over from hg19 to hg38. Score cut-offs of ≥3 or ≥5 were applied to the final output in order to determine which

bait–target pairs were considered as interacting, with the rest assigned as non-interacting.

Binned 10-kb regions were overlapped with Capture HIC protocol defined baits[12], lifted over to hg38. For each such 'bait bin', potential targets within 2 Mb (200 bins) were assigned and their numbers reduced according to the presence of CAGE tags in more than one replicate in at least one of the ENCODE cell lines. For analyses based on enhancer-associated targets, targets were considered based on overlap with at least one CAGE-defined enhancer which was active (more than one tag in more than one replicate) in at least one of the ENCODE cell lines. Only bait–target pairs which fell into the distance range of between 6 and 200 bins were considered (corresponding to >50 kb and up to 2 Mb).

The full list of features considered in the interaction modelling are listed in Supplementary Table 3. Features from the PI and PD profiles were added for each bait–target pair separately for the bait and the target (see Supplementary Table 2 for their descriptions). Enhancer information was added for the bait and targets based on the FANTOM5 enhancer set for hg38 (see above)—including the total eRNA output produced at the enhancer (replicate sum), the number of enhancers deemed to be active within the bin in the given cell type (at least one tag in at least one replicate) and the number of cell lines supporting the target enhancer (number of ENCODE cell lines with at least one enhancer active within the bin). Bin directionality was calculated based on pooled replicates, using a transcriptional directionality score[31] and assigning a value of 0 where no tags were present. The boundary insulation between the bait and the target was calculated according to the number of XAD boundaries observed in the intervening bins. Boundary insulation was defined up to 3; all values >3 were given a score of 3. The peaks detected were the number of CAGE peaks overlapping the bait or target bin, according to the hg38 DPI set from FANTOM5, which was transcribed in that cell type (at least one tag in at least one replicate). The cross-correlation for the PI and PD components was calculated per chromosome by correlating the respective PI and PD profiles over four ENCODE cell lines between the bait and target bins, specifying Kendall for the correlation method. The first eigenvector was calculated from the cross-correlation matrices for each chromosome and its value was supplied for the bait and target bins. Distance was defined as the number of 10-kb bins separating the bait and the target.

To assess model performance, two data sets from GM12878 were analysed. For assessing the applicability of the model trained in GM12878 for predicting interactions in other cell types, the data set as generated was kept in its current format (termed 'raw' format). In addition to this, since the number of positive interactions decayed sharply with larger distances, a second data set was generated where the ratio of negative-to-positive cases over distance was balanced by randomly sampling 20 negatives to each positive in the data set, with replacement to account for cases where the negative rows were not at least 20-fold in number to the positive rows.

To generate a balanced data set for training, SMOTE[44], as implemented in the unbalanced package (version 2.0) in R, was applied to the data, specifying parameters percOver = 200 and percUnder = 150 to generate new positives together with under-sampling the negatives to achieve a balance of 1:1 in the data set. In order to balance the data set most fairly over distances, SMOTE was applied separately across each possible bait–target separation.

The model training, predictions and validations were performed in R using the randomForest package supplemented with foreach (version 1.4.3) and doParallel (version 1.0.10) to run on multiple cores. A 10-fold cross-validation was used, whereby the data set was split randomly into 10 equal-sized pieces and a single piece was held out as a testing set on each of the 10 runs of the model which was trained on the remaining 90%. All training was carried out based on a ChiCAGO score cut-off of 3. All performance statistics and probability estimates in GM12878 were based on predictions made across the held-out runs over the full data set.

To assess whether there are different features which are important for specific distances without the bias of most examples being weighted towards short distances, the above analysis outline was repeated for the same data set but restricted to three possible distances: (50 kb, 250 kb), (250 kb, 500 kb) and (500 kb, 2 Mb). For each set of distances, the mean decrease accuracy (MDA) was averaged across the 10 runs in order to obtain a final feature performance.

To find the optimal probability cut-off for calling a predicted interaction, the value for which the F1 statistic was maximised was calculated using the optim package in R, according to the desired score cut-off. Since the most efficient cut-off is not fixed according to distance, the F1 statistic was maximised separately for five sets of bait–target distances: (50 kb, 100 kb), (100 kb, 250 kb), (250 kb, 500 kb), (500 kb, 1 Mb) and (1 Mb, 2 Mb), and performance was analysed for predictions generated using the resulting cut-offs. To calculate the effect of the CHiCAGO score cut-off on precision and recall, we optimised the F1 statistic separately for the five sets based on a range of score cut-offs (0.1, 0.25, 0.5, 1, 1.5, 2, 2.5, 3, 3.5, 4, 4.5, 5, 5.5 and 6).

The pROC[71] package in R was used to generate AUC statistics and the caret package (version 6.0-73) in R was used to generate the precision, recall and F1 statistics. Plots were generated using custom functions based on statistics generated from the ROCR[67] package.

The data set for GM12878 was trained as described above against a score of ≥3 and was used to predict on the equivalent set of bin pairs in HeLa-S3 and HepG2, which were subsequently reduced to those with enhancer targets only (using the same criteria as described above). Final probabilities were calculated based on the

mean of probabilities over the 10 runs. The distance-based F1-maximising cut-offs were applied to obtain a final list of interactions. EP interaction sharing between GM12878, HeLa-S3 and HepG2 was calculated based on whether the interactions were present in 1, 2 or 3 of the cell types and venn diagrams were generated using the R package VennDiagram (version 1.6.17).

Processed Pol II ChIA-PET interaction data for HeLa-S3 were downloaded from ENCODE[38] and start and end regions were lifted over to hg38, removing those without a 1:1 correspondence between the two builds. Bait–target pairs for predicted EP interactions in HeLa-S3 were selected according to a probability of at least 0.6 in HeLa-S3 and a probability of less than 0.4 in GM12878. Both ends were intersected with the lifted over start and end regions in the ChIA-PET data in order to generate a list of candidate examples. We selected an example on chromosome 6 due to that chromosome's high-performance statistics from the modelling described above. The R package Sushi (version 1.10.0) was used to generate plots of the resulting loops and lines together with annotations and enhancer locations for the data sets in the example region.

**Analysis and predictions across 76 cell types**. A total of 249 FANTOM5 CAGE libraries were selected according to the availability of sample replicates. Transcriptional decomposition was applied to generate PD and PI components for a total of 76 cell types (including four ENCODE cell lines and 72 primary cells, see Supplementary Data 6 for a list of library identifiers and names). For the purposes of consistency between the ENCODE-generated data sets and the primary cell-type-generated data sets, the hyperparameters were fixed for the random walk and independent components to the same values which were generated from the models in ENCODE (while allowing the hyperparameters for the zero-inflated negative binomial distribution to vary).

Raw binned data at 10-kb resolution were normalised into tags per million (TPM) and the mean was taken across replicates to obtain a matrix of 76 columns against the total genomic bin count (303,065). Only regions potentially transcribed in the given set of CAGE libraries were considered by asking for bins which had more than one cell type containing tags. The matrix was transformed into $\log_{10}$ values (adding a pseudo-count of 1) and hierarchical clustering using hclust (complete linkage method) was applied in R. The ordering from the clustering was used to guide the row and column ordering in the heatmaps for Fig. 6. The R function cutree was used to find 10 groups from the clustering, which were annotated and merged manually to generate the most biologically relevant cell-type groupings from the data.

A common set of bins (34,953) was derived for comparison between cell types of the PD and the PI components by choosing the set of bins where the sign of the PI component was positive in more than one cell type. To generate similarity matrices, the PD and PI signals were converted into binary values according to whether the signal was positive (1) or negative (0) (PI–0.1 was used for the independent component to avoid non-expressed bins with very small positive estimates). The resultant matrix of 76 columns and rows according to the common bin set was used to calculate $1-L_1$ norm between each pair of cell types to calculate a similarity matrix. The same metric was used for calculating similarity matrices representing cell-type boundary sharing and cell-type EP sharing, with methods described in the sections below. The R package pheatmap (version 1.0.8) was used to generate cell-type group-annotated heatmaps directly from the similarity matrices, using the cell-type ordering from the hierarchical clustering in the raw data.

For all cell types, XAD boundaries were calculated from the algorithm described above for ENCODE data sets, supplying the stability scores across the full set of cell types. To calculate boundary sharing across all cell types, non-overlapping bins of size of 100 kb were generated and calculated for each expanded bin in the number of cell types within which a boundary was found. In order to cover all possible windows, the expanded bins were shifted by a 10-kb bin 10 times and the final number of shared boundaries was calculated according to an average of 10.

Data sets with features were generated as described for the ENCODE cell lines for all cell types, with the bin selection also extended to the full set, thus creating a common bin set which is larger than that for the ENCODE cell lines alone. Features non-specific to a cell type were also calculated more broadly with consideration to the 76 cell types. The model was trained as above using the GM12878 data set and the broader bin set. Similar model performance was noted or this model when testing on the raw data set using 10-fold cross-validation. The held-out set predicted probabilities were used for GM12878 and the predicted probabilities for the other 75 cell types were generated by averaging over 10 trained models across the whole data set (to robustly account for random differences in the data balancing for the training).

The data sets for the 76 cell types were reduced according to whether at least one data set had an active CAGE enhancer annotated to it, in order to obtain a list of potential EP interactions. To generate lists of predicted interactions, we generated distance-based probability cut-offs in GM12878, using the same method for the ENCODE data sets above, using score cut-offs of ≥3 and ≥5.

To calculate EP interaction-sharing statistics, the ≥3-score cut-off was used and for each possible number of cell types (from 1 to 76), the number of significant interactions which were present in exactly that number of cell types was calculated. To generate the EP-sharing heatmap, all interactions which were present in more than one cell type were considered and pairwise cell-type similarity was calculated from $(1-L_1$ norm) between columns of the binary (1 predicted/0 not predicted)

matrix with cell types as columns and interaction set as rows. This generated a similarity matrix which was plotted using the R package pheatmap (version 1.0.8) based on the cell-type ordering from the hierarchical clustering in the raw data.

To isolate examples of interactions specific to blood, interactions meeting the criteria of present (≥3-score distance-based efficient cut-offs) in at least half of the cell types labelled as 'blood cells' (Supplementary Data 6) and in fewer than a total of up to a maximum of three more than the number of blood cells within the 76 cell types were classified as 'blood specific'. For further trimming of examples and plotting of loops, probabilities assigned to all cell types assigned as blood cells were averaged into one using the mean. Merged probability vectors were also generated for endothelial cells, epithelial cells, muscle cells, mesenchymal cells and fibroblasts. Examples of blood-specific interactions were selected based on the loop being still significant according to the averaged probabilities. The R package Sushi (version 1.10.0) was used to generate plots of selected examples, including tracks for annotations and the averaged PD, PI components for blood cells.

GWAS trait-associated SNPs and those in LD were retrieved from the traseR R package. SNP hg19 coordinates were lifted to hg38. For each trait with at least 50 assigned SNPs, an $\chi^2$ test was constructed per each of the 76 cell types and components (PD, PI and XAD boundaries EP interactions). For the PD and PI components, the foreground was assigned as the set of bins with a positive component sign in the given cell type, and the non-foreground was assigned as the set of bins with a positive component sign in at least one cell type, but not the given cell type. For the EP interactions, the foreground was assigned as the set of target enhancers within the given cell type and the non-foreground based on the target enhancers in any other cell type. For the XAD boundaries, the foreground was assigned as the boundary bins ±2 bins in the given cell type against a non-foreground of all boundary bins ±2 bins in any cell type, not including the given cell type. The odds ratios were calculated based on $2 \times 2$ contingency tables including the number of trait SNPs within the foreground/non-foreground and the number of trait SNPs for all other trait SNPs within the foreground/non-foreground. For each component, the p values from the total set of $\chi^2$ tests were corrected to an FDR with the Benjamini–Hochberg correction.

The R package pheatmap (version 1.0.8) was used to generate heatmaps of the resulting trait enrichments and the R package Sushi (version 1.10.0) was used to generate interaction plots for the given EP interaction examples. To generate the heatmaps for the number of cell types with significant trait enrichments, the number of cell types with FDR<0.01 and an odds of >1.25 were counted per trait, per component. To calculate the component preference by trait, the resulting data were normalised per trait before generating the heatmap.

Values for the PD and PI components, together with locations of XAD boundaries and predicted EP interactions were saved out as BED files, using UCSC zero-based coordinates, and supplied as supplementary files at Zenodo (https://doi.org/10.5281/zenodo.556727, Rennie et al.[72]).

**Code availability**. The code for transcriptional decomposition and downstream analyses is available at https://github.com/anderssonlab/transcriptional_decomposition.

**Data availability**. Transcriptional decomposition data and downstream results are available at https://doi.org/10.5281/zenodo.556727 (Rennie et al.[72]). FANTOM5 hg38 enhancers are available at https://doi.org/10.5281/zenodo.556775 (Dalby et al.[43]).

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

## Acknowledgements

This project has received funding from the Danish Council for Independent Research (Grant 6108-00038) and the European Research Council (ERC) under the European

Union's Horizon 2020 research and innovation programme (Grant 638173). We would like to thank Hideya Kawaji for remapping and DPI tag clustering of FANTOM5 CAGE data. Furthermore, we thank Alvaro Rada-Iglesias and members of the Robin Andersson lab and Albin Sandelin lab for their rewarding comments and discussions.

## Author contributions

S.R and R.A. conceived the project; S.R conducted most analyses, with support from M. D., L.v.D. and R.A., S.R. and R.A wrote the paper; all authors reviewed the final manuscript.

## Additional information

**Competing interests:** The authors declare no competing financial interests.

