## [Peer Review File · Nature Communications]

Reviewers' comments:

Reviewer #1 (Remarks to the Author):

In this work, Renie and colleagues use a decomposition approach to mine additional information from gene expression data in order to infer spatial chromatin relationship in a cell type-specific manner. Their approach is novel and interesting, comes with particular caveates that are openly discussed (e.g., does not adequately explain poorly-expressed genomic regions and their folding), and offers a new way for investigating potentially important interactions with regulatory elements carrying genetic variants linked to disease phenotypes. Overall, this is a timely and interesting study, there is rich scope in endorsing its publication, and I offer some points for its improvement below.

Specific comments:

1. The whole computational approach, in the main text, should be explained in more detail and in a way that allows readers of a mostly non-bioinformatic background to come to terms with the notion behind the algorithm.
2. In Fig. 2, PD allows for the discrimination of the inactive versus the active state of two exemplary genes between three cell types. This raises two questions; first, how many such genes does the PD component allow us to detect (out of the whole gene expression repertoire)? What fraction of the non-/poorly-expressed genes do these represent? And, could the authors explain, why do XADs essentially only predict "active" domains although the PD component also distinguishes inactivity (to some extent)?
3. In Fig. 3C,D it is not at all clear to this reviewer what the different elements that this differential analysis are and how they differ when teasing the PD or the PI components? Is there something that we can learn from such an analysis besides the "predictability" of each component?
4. Fig. 3E shows some differential enrichment (not sure about the scale/units of the heatmap here) of TFs recognition motifs, but we lack a number of information to make this useful. For example, how many of the mentioned TFs are actually expressed per cell type examined? How are they related to the core transcription program of the cell type and/or of the gene subset studied? Also, we know well that the presence of a motif does not ensure TF binding (e.g., NFkappaB has more than >0.5 cognate sites across the genome, but binds ~8,000). My suggestion is to move this panel to the supplement of the manuscript.
5. In Fig. 4, it would be informative to see the average offset of XAD boundaries to their cognate TADs, although the authors do indeed mention that XADs are "proximal" to TADs.
6. It is not clear to this reviewer why a 20:1 NP ratio was used to train the algorithm. I understand this proved optimal, but how is this explained and how does a less unbalanced ratio would impact the prediction should be discussed.
7. Fig. 6A-C, it appears that the PI-derived heatmaps display most dynamic range and mostly resemble E-P interactions here, although the majority of predictions before were based on PD. Is this so? And, if it is so, how do the authors explain this? Does this perhaps relate to the putative differential TF binding?
8. Overall, the authors' analyses point to an important contribution of transcription in genomic spatial organization -- under this notion, there should be some discussion of the "transcription factory" model for E-P interactions (Papantonis & Cook, Chem Rev, 2013; Papantonis et al, EMBO J, 2012), and of the "A/B compartment" (Lieberman-Aiden et al, Science, 2009) significance in larger-scale spatial positioning preferences (which also begs the question whether A/B

compartments can be reconstructed using this approach?).

Reviewer #2 (Remarks to the Author):

The manuscript by Rennie and colleagues, "Transcriptional decomposition reveals active chromatin architectures and cell specific regulatory interactions", describes a creative new computational approach to decompose gene expression into separate "regional" (position dependent, PD) and "local" (position independent, PI) components. The authors developed an elegant statistical model to decompose PD and PI from biological replicates of CAGE and RNA-seq data. The boundaries of regions high in the PD component were observed to correlate with topological associated domain boundaries and to chromatin modifications, providing a link between these poorly understood marks and their effects on gene expression.

In general this is a creative, novel, and interesting paper that should be published. I do, however, have several comments that should be addressed prior to publication.

Comments:

(1) A reasonable interpretation, which is supported by many of the author's observations, is that the PD component reflects the regional effects of chromatin on gene expression, for instance PRC2-mediated silencing. However, the authors primarily motivate the decomposition by pointing to recent work on topological domains/ 3-dimensional chromatin interactions found using Hi-C. Although both aspects of nuclear biology might undoubtedly play a role, which of these distinct representations are causal remains unclear. The focus on Hi-C data/ chromatin interactions is trendier, and I can understand the author's rationale for putting their work in this context. In my opinion, I think the manuscript would read better if the authors worked regional chromatin effects into their rationale more seamlessly in the introduction (though I will also note here that it is entirely up to the authors how they would like to introduce their work).

(2) My understanding of the decomposition for PI and PD components is that it depends heavily on having substantial biological noise in gene expression between replicates of the same cell type. I wonder how the method would perform if it were given technical replicates with no systematic biological variation?! Some discussion of how the authors think about this would be useful for readers.

(3) Presumably there may be systematic biases distributed across the genome (isochores of GC content, for instance) that could cause systematic technical noise over the genome, and lead to misleading PD component signals. These types of potential confounding signals do not appear to be discussed or controlled for in the current manuscript. Providing the reader a window into the effects of these types of potential technical biases in their discussion, and/ or (even better) fitting the data to technical replicates or aligning on GC isochores would be a potentially useful control that would help to eliminate alternative interpretations.

(4) Would the model provide even more information in cases where there is additional biological variation in the data? Analyzing the same cell types among many different human genotypes (available in the GTEx RNA-seq datasets) might allow the detection of XADs with higher resolution/ accuracy.

Reviewer #3 (Remarks to the Author):

This study analyzed RNA expression data to explain chromatin compartments and interactions including enhancer-promoter interactions using statistical modeling and machine learning. The authors decomposed expression data into position -dependent or -independent components.

Positional information content in PD components was shown by the comparison with histone modifications and with HiC-derived compartments. Furthermore, features derived from the expression data were used to model XAD and EP interactions. The models built in the study were applied to other cell lines and showed cell type specificity of EP interactions. Potential application to disease associated genetic variants was demonstrated.

Overall, the study is well structured to show that expression values can be dissected, and can be utilized to predict XAD and EP interactions. Many studies modeled EP pairing from integrating expression data, epigenetic markers and interaction data. However, it is novel that chromatin compartmentalization and interactions can be derived from expression values without epigenetic information. The ability to predict the chromatin compartments and EP interactions from RNA expression data only, one of the most commonly practiced genome wide experiments, would be a valuable asset to many researchers.

The study can be considered for publication after the following specific concerns have been addressed.

1. Figure 2B compares the signals of PD components with H3K27me3 and H3K36me3. Presenting signals of PI components will help to demonstrate PD components hold the information on chromatin compartments, but not PI components.

2. In Figure 3B, the authors used DNaseI, H2A.Z, H3K4me3 and H3K27Ac to support that PI components contain promoter localized information. Although active regions show high signals of these datasets, they do not exclusively represent promoters. In order to strengthen the arguments, one could predict the presence of H3K4me1, annotated promoters or distance to known TSS.

Plots to compare PI and PD on known promoters and enhancers would also be helpful. It was mentioned in the discussion that enhancer regulation is explained in PI components (Discussion; paragraph4; last sentence). This statement must be supported. Can PI components predict not only promoters but active regulatory elements?

3. 10kb bin is used for EP interactions. Is the resolution enough to differentiate distinct regulatory elements? How is it dealt with in cases where there are more than two regulatory elements in one bin?

4. PI components have promoter information (Figure 3B), enriched motifs of cell type specific transcription factors (Figure 3E), and show cell type specific expression level (Figure 6B). On the other hand, PD components carry less information on cell type specificity (Figure 1A text, Figure 6A). Does this mean that the cell type specific interactions are more explained by PI components, despite that PD is the most important feature to predict the (cell type specific) regulatory interactions (Figure 5B)? This must be clarified.

5. Fig. S4C labels do not match with Suppl. Table S4.

Response to Reviewers' comments:

Reviewer #1 (Remarks to the Author):

In this work, Renie and colleagues use a decomposition approach to mine additional information from gene expression data in order to infer spatial chromatin relationship in a cell type-specific manner. Their approach is novel and interesting, comes with particular caveates that are openly discussed (e.g., does not adequately explain poorly-expressed genomic regions and their folding), and offers a new way for investigating potentially important interactions with regulatory elements carrying genetic variants linked to disease phenotypes. Overall, this is a timely and interesting study, there is rich scope in endorsing its publication, and I offer some points for its improvement below.

Specific comments:

1. The whole computational approach, in the main text, should be explained in more detail and in a way that allows readers of a mostly non-bioinformatic background to come to terms with the notion behind the algorithm.

We appreciate the feedback and agree that the manuscript will benefit from a better explanation of the computational approach. In the revised manuscript, we have expanded the description of the transcriptional decomposition approach, including a less technical interpretation of the approach.

2. In Fig. 2, PD allows for the discrimination of the inactive versus the active state of two exemplary genes between three cell types. This raises two questions; first, how many such genes does the PD component allows us to detect (out of the whole gene expression repertoire)? What fraction of the non-/poorly-expressed genes do these represent?

Regions of positive PD signal were highly enriched in expressed genes within a given cell type (odds ratio ranging from 4.3 to 15.9, $p < 2.2e-16$ for ENCODE tier 1 cell lines, Fisher's exact test), with on average 93% of expressed genes located in positive PD regions. These results are now included in the revised manuscript.

And, could the authors explain, why do XADs essentially only predict "active" domains although the PD component also distinguishes inactivity (to some extent)?

Our analysis shows that changes in the PD component are strongest at boundaries of active regions. Therefore, we argue that XADs better reflect active domains. While inactive regions could be defined as XADs in between active XADs, we will likely not capture sub-structures within larger regions of inactive chromatin due to low or lack of transcriptional activity, which our modelling relies on. In the revised manuscript, we have added a sentence explaining the low overlap between XAD boundaries and boundaries of inactive TADs.

3. In Fig. 3C,D it is not at all clear to this reviewer what the different elements that this differential analysis are and how they differ when testing the PD or the PI components? Is there something that we can learn from such an analysis besides the "predictability" of each component?

We thank the reviewer for pointing out that this was not explained well in the text. Figures 3C,D show differential expression p-value (FDR adjusted) versus fold change when comparing PD component expression data (posterior estimates) between expressed (≥ 10 CAGE reads) GM12878 and HeLa-S3 genomic bins. An explanation of the test has been added in the revised manuscript.

4. Fig. 3E shows some differential enrichment (not sure about the scale/units of the heatmap here) of TFs recognition motifs, but we lack a number of information to make this useful. For example, how many of the mentioned TFs are actually expressed per cell type examined? How are they related to the core transcription program of the cell type and/or of the gene subset studied? Also, we know well that the presence of a motif does not ensure TF binding (e.g., NFkappaB has more than >0.5 cognate sites across the genome, but binds $\sim 8,000$). My suggestion is to move this panel to the supplement of the manuscript.

We thank the reviewer for the feedback and for the suggestion to consider TF expression in the analysis. We have now analysed the TF enrichment taking into account the TF expression levels in the cell types considered, using RTFBSDB method (doi: 10.1093/bioinformatics/btw338). Our new results show more specific enrichments of cell-type related TFs. While we agree with the reviewer that TF motif enrichments should not be over-interpreted, even when TF expression is taken into account, we believe that the analysis provides insights and helps in interpreting the differences between transcriptional components. Therefore, we suggest keeping the new heatmap in figure 3 but would be willing to only have the results in the supplementary if needed.

5. In Fig. 4, it would be informative to see the average offset of XAD boundaries to their cognate TADs, although the authors do indeed mention that XADs are "proximal" to TADs.

We thank the reviewer for the suggestion. In the revised manuscript, Supplementary Figure S4 now includes a density of the distances between XAD boundaries and TAD boundaries in GM12878 cells.

6. It is not clear to this reviewer why a 20:1 NP ratio was used to train the algorithm. I understand this proved optimal, but how is this explained and how does a less unbalanced ratio would impact the prediction should be discussed.

We apologies for not being clear enough in the text. The original text read:

“In each cross-validation round, we balanced the training data and predicted on held-out data at 20:1 or raw (unmodified) negative:positive (NP) ratios.”

All training was done on balanced data. Predictions on held-out data was evaluated at a 20:1 as well as raw negative:positive ratios. We trained on balanced data in order to not make the dominant class (negatives) take over in the model learning. But, it is important to evaluate the predictive performance on real negative:positive ratios. The 20:1 ratio was also included in order to relate to a previous method for enhancer-promoter interactions (Whalen et al. Nat Gen 2016). We have tried to make this clearer in the revised manuscript.

7. Fig. 6A-C, it appears that the PI-derived heatmaps display most dynamic range and mostly resemble E-P interactions here, although the majority of predictions before were based on PD. Is this so? And, if it is so, how do the authors explain this? Does this perhaps relate to the putative differential TF binding?

The PI is showing larger differences between cell types than the PD. This is expected, since we expect the PD component to reflect more stable domain architectures and the PI component to reflect expression-associated changes. The PI component is indeed affecting the predictions of E-P interactions as shown by Fig. 5C. However, we show that the PD structure is highly informative of interactions, especially the proximal ones, while expression levels (better reflected by the PI component) is more important for predicting long-range interactions (Fig. 5C-E). In the revised manuscript, we have extended the discussion section to better cover these aspects.

8. Overall, the authors' analyses point to an important contribution of transcription in genomic spatial organization -- under this notion, there should be some discussion of the "transcription factory" model for E-P interactions (Papantonis & Cook, Chem Rev, 2013; Papantonis et al, EMBO J, 2012), and of the "A/B compartment" (Lieberman-Aiden et al,

Science, 2009) significance in larger-scale spatial positioning preferences (which also begs the question whether A/B compartments can be reconstructed using this approach?).

We thank the reviewer for recognising the value of our contribution and for this suggestion. Indeed, we show that the PD component but not the PI component is very informative of A/B compartments (Fig. 3A and Supp. Fig. S1). Models of RNAPII-centred chromatin architectures are very relevant to discuss in relation to our findings. We have extended the Discussion section to this end in the revised manuscript.

Reviewer #2 (Remarks to the Author):

The manuscript by Rennie and colleagues, “Transcriptional decomposition reveals active chromatin architectures and cell specific regulatory interactions”, describes a creative new computational approach to decompose gene expression into separate “regional” (position dependent, PD) and “local” (position independent, PI) components. The authors developed an elegant statistical model to decompose PD and PI from biological replicates of CAGE and RNA-seq data. The boundaries of regions high in the PD component were observed to correlate with topological associated domain boundaries and to chromatin modifications, providing a link between these poorly understood marks and their effects on gene expression.

In general this is a creative, novel, and interesting paper that should be published. I do, however, have several comments that should be addressed prior to publication.

Comments:

(1) A reasonable interpretation, which is supported by many of the author’s observations, is that the PD component reflects the regional effects of chromatin on gene expression, for instance PRC2-mediated silencing. However, the authors primarily motivate the decomposition by pointing to recent work on topological domains/ 3-dimensional chromatin interactions found using Hi-C. Although both aspects of nuclear biology might undoubtedly play a role, which of these distinct representations are causal remains unclear. The focus on Hi-C data/ chromatin interactions is trendier, and I can understand the author’s rationale for putting their work in this context. In my opinion, I think the manuscript would read better if the authors worked regional chromatin effects into their rationale more seamlessly in the introduction (though I will also note here that it is entirely up to the authors how they would like to introduce their work).

We appreciate the feedback from the reviewer. We do think that the overall observations on how positionally attributable vs independent expression fits into three dimensional chromatin organisations are also applicable to large-scale chromatin effects. This is partly because the two (regional chromatin effects and three-dimensional organisation) are related, with domain boundaries co-occurring with changes in large-scale chromatin silencing/activation. In the revised manuscript, we have added the aspect of regional chromatin effects to the introduction.

(2) My understanding of the decomposition for PI and PD components is that it depends heavily on having substantial biological noise in gene expression between replicates of the same cell type. I wonder how the method would perform if it were given technical replicates with no systematic biological variation?! Some discussion of how the authors think about this

would be useful for readers.

We thank the reviewer for considering scenarios in which the transcriptional decomposition might not work well. However, we think the reviewer has misinterpreted the approach. It is correct that replicate measurements are important for the modelling in order to provide accurate posterior estimates of the transcriptional components and to identify regions of confident posterior samplings of the decomposed expression values. This is apparent for XAD boundary calling, for which both posterior standard deviations within a cell type and their stabilities across cell types is a good predictor. However, substantial biological noise between replicates would likely make posterior estimates of transcriptional components less accurate for the cell type. Across samples, concurrent modelling (as done for the ENCODE tier 1 cell lines) could be beneficial in model fitting, as this assumes a common distribution for the hyperparameters. Note that, in such a case, the different samples are not considered to be replicates.

(3) Presumably there may be systematic biases distributed across the genome (isochores of GC content, for instance) that could cause systematic technical noise over the genome, and lead to misleading PD component signals. These types of potential confounding signals do not appear to be discussed or controlled for in the current manuscript. Providing the reader a window into the effects of these types of potential technical biases in their discussion, and/ or (even better) fitting the data to technical replicates or aligning on GC isochores would be a potentially useful control that would help to eliminate alternative interpretations.

We thank the reviewer for raising these concerns. We are aware that systematic biases could influence our results, as is the case for all sequencing-based genomics investigations. However, these biases are not trivial to account for. For instance, although it is known that GC-content can affect sequencing results, this is confounded with the general preference of transcription start sites at CpG island promoters, which is reflected in CAGE data. Thus, it is not clear how such potential biases should be dealt with. An attempt to remove such biases (e.g. reflecting GC content) could therefore likely remove true biological signal. A systematic investigation of such biases and how to deal with these in CAGE data would be important but we believe such an undertaking would be out of scope in this study.

(4) Would the model provide even more information in cases where there is additional biological variation in the data? Analyzing the same cell types among many different human genotypes (available in the GTEx RNA-seq datasets) might allow the detection of XADs with higher resolution/ accuracy.

We agree with the reviewer that applying the transcriptional decomposition approach across individuals with different genotypes would be very interesting and are investigating that

aspect in a separate project. However, we do believe that such an analysis warrants for a whole paper on its own and is out of scope in this manuscript.

Reviewer #3 (Remarks to the Author):

This study analyzed RNA expression data to explain chromatin compartments and interactions including enhancer-promoter interactions using statistical modeling and machine learning. The authors decomposed expression data into position -dependent or -independent components. Positional information content in PD components was shown by the comparison with histone modifications and with HiC-derived compartments. Furthermore, features derived from the expression data were used to model XAD and EP interactions. The models built in the study were applied to other cell lines and showed cell type specificity of EP interactions. Potential application to disease associated genetic variants was demonstrated.

Overall, the study is well structured to shows that expression values can be dissected, and can be utilized to predict XAD and EP interactions. Many studies modeled EP pairing from integrating expression data, epigenetic markers and interaction data. However, it is novel that chromatin compartmentalization and interactions can be derived from expression values without epigenetic information. The ability to predict the chromatin compartments and EP interactions from RNA expression data only, one of the most commonly practiced genome wide experiments, would be a valuable asset to many researchers.

The study can be considered for publication after the following specific concerns have been addressed.

1. Figure 2B compares the signals of PD components with H3K27me3 and H3K36me3. Presenting signals of PI components will help to demonstrate PD components hold the information on chromatin compartments, but not PI components.

We thank the reviewer for the suggestion. We have added the PI signal to Fig. 2B,C in the revised manuscript.

2. In Figure 3B, the authors used DNaseI, H2A.Z, H3K4me3 and H3K27Ac to support that PI components contain promoter localized information. Although active regions show high signals of these datasets, they do not exclusively represent promoters. In order to strengthen the arguments, one could predict the presence of H3K4me1, annotated promoters or distance to known TSS.

Plots to compare PI and PD on known promoters and enhancers would also be helpful. It was mentioned in the discussion that enhancer regulation is explained in PI components (Discussion; paragraph4; last sentence). This statement must be supported. Can PI components predict not only promoters but active regulatory elements?

Our rationale for Fig. 3B was to investigate the coupling between PD and PI components to expression-associated chromatin marks. We haven't claimed that PD or PI components could be used to distinguish between or identify active enhancers or promoters. We argue that the presence of localised divergent transcription analysed from non-decomposed CAGE data is better suited for this purpose. However, we make use of both localised enhancer and promoter expression levels and features derived from transcriptional components to model enhancer-promoter interactions. In the revised manuscript, we have tried to better explain how these aspects individually contribute to predictions of enhancer-promoter interactions.

3. 10kb bin is used for EP interactions. Is the resolution enough to differentiate distinct regulatory elements? How is it dealt with in cases where there are more than two regulatory elements in one bin?

It is true that in the E-P interaction modelling we are training on 10kb resolution Hi-C and decomposed data, but we also include non-decomposed expression of local FANTOM enhancers and promoters, as described above. All potential pairs of enhancers and promoters between considered bins were evaluated. It is a valid concern that 10kb might be too low resolution to distinguish between interactions in some regions dense with regulatory elements. The reason why we mapped HiC data to this resolution is that we wanted to be able to directly compare with our transcriptional decomposition results (same resolution). In the revised manuscript, we have stated this argument directly.

4. PI components have promoter information (Figure 3B), enriched motifs of cell type specific transcription factors (Figure 3E), and show cell type specific expression level (Figure 6B). On the other hand, PD components carry less information on cell type specificity (Figure 1A text, Figure 6A). Does this mean that the cell type specific interactions are more explained by PI components, despite that PD is the most important feature to predict the (cell type specific) regulatory interactions (Figure 5B)? This must be clarified.

We apologise for the confusion and have tried to better clarify the differences between PD and PI components regarding cell type-specificity and their importance in E-P interaction modelling. The PI component is affecting the predictions of E-P interactions as shown by Fig. 5C. However, we show that the PD structure is highly informative of interactions especially the proximal ones, while expression levels (better reflected by the PI component) are more important for predicting long-range interactions (Fig. 5D-E). In the revised manuscript, we have extended the discussion section to better cover these aspects.

5. Fig. S4C labels do not match with Suppl. Table S4.

We thank the reviewer for identifying these discrepancies. The labels have been corrected.

Reviewers' comments:

Reviewer #1 (Remarks to the Author):

I have read the responses of the authors to the comments of all three reviewers (which were anyhow converging), and I am fully satisfied by them. In my opinion all concerns have been dealt with adequately, and the manuscript can be published in its current form.

Reviewer #2 (Remarks to the Author):

My comments have been addressed by the authors.

Reviewer #3 (Remarks to the Author):

The authors responded to the concerns raised and made the necessary changes to the manuscript. However, there is still a remaining concern that must be addressed. After that I support acceptance.

In the response to the second question (originally read as below), the authors stated that PI components predict expression-associated chromatin marks, which are not only promoter-associated features but regulatory element associated features. This explains better than the current main text, stating the tested chromatin marks are "associated with features of (transcriptionally) active promoters", and "PI component, in contrast to the PD component, carries information about promoter-localised" effects.

I suggest the authors can change the main text to state these marks test are associated with transcriptionally active regulatory elements (not only with active promoters).

2. In Figure 3B, the authors used DNaseI, H2A.Z, H3K4me3 and H3K27Ac to support that PI components contain promoter localized information. Although active regions show high signals of these datasets, they do not exclusively represent promoters. In order to strengthen the arguments, one could predict the presence of H3K4me1, annotated promoters or distance to known TSS. Plots to compare PI and PD on known promoters and enhancers would also be helpful. It was mentioned in the discussion that enhancer regulation is explained in PI components (Discussion; paragraph4; last sentence). This statement must be supported. Can PI components predict not only promoters but active regulatory elements?

Response to Reviewers' comments:

Reviewer #3 (Remarks to the Author):

The authors responded to the concerns raised and made the necessary changes to the manuscript. However, there is still a remaining concern that must be addressed. After that I support acceptance.

In the response to the second question (originally read as below), the authors stated that PI components predict expression-associated chromatin marks, which are not only promoter-associated features but regulatory element associated features. This explains better than the current main text, stating the tested chromatin marks are "associated with features of (transcriptionally) active promoters", and "PI component, in contrast to the PD component, carries information about promoter-localised" effects.

I suggest the authors can change the main text to state these marks test are associated with transcriptionally active regulatory elements (not only with active promoters).

2. In Figure 3B, the authors used DNaseI, H2A.Z, H3K4me3 and H3K27Ac to support that PI components contain promoter localized information. Although active regions show high signals of these datasets, they do not exclusively represent promoters. In order to strengthen the arguments, one could predict the presence of H3K4me1, annotated promoters or distance to known TSS. Plots to compare PI and PD on known promoters and enhancers would also be helpful. It was mentioned in the discussion that enhancer regulation is explained in PI components (Discussion; paragraph4; last sentence). This statement must be supported. Can PI components predict not only promoters but active regulatory elements?

We thank the reviewer for this suggestion. In our original text we did not mean to make any distinction between gene-promoters and gene-distal regulatory elements with promoter activity. Nevertheless, we agree with the reviewer that a more accurate conclusion is to associate PI with transcriptionally active regulatory elements. We have therefore revised the paragraph:

"To examine the localised patterns observed for expression levels not attributable to position (PI component, Fig. 2A), we trained a random forest model (Methods) on GM12878 transcriptional components to predict the presence or absence of DNase I hypersensitive sites (DHSs), histone variant H2A.Z and post-translational histone modifications H3K4me3 and H3K27ac (binarised DNase-seq and ChIP-seq data³⁷ in each bin), each associated with features of (transcriptionally) active regulatory elements³⁸. The resulting models allowed us to generate a probability distribution for each mark given each transcriptional component. For all tested marks we observed a clear bias with stronger predictive power from the PI component than the PD component (Fig. 3B). These results indicate that the PI component, in contrast to the PD component, carries information about regulatory element-localised and expression-level associated effects."

REVIEWERS' COMMENTS:

Reviewer #3 (Remarks to the Author):

The authors have successfully address my additional remaining concern and the manuscript can be accepted.